# SelfCodeAlign: Self-Alignment for Code Generation

**Yuxiang Wei**[1] **Federico Cassano**[2,7] **Jiawei Liu**[1] **Yifeng Ding**[1] **Naman Jain**[3] **Zachary Mueller**[5]
**Harm de Vries**[4] **Leandro von Werra**[5] **Arjun Guha**[2,6] **Lingming Zhang**[1]

[1]University of Illinois Urbana-Champaign    [2]Northeastern University
[3]University of California, Berkeley   [4]ServiceNow Research   [5]Hugging Face   [6]Roblox   [7]Cursor AI

✉ {ywei40,lingming}@illinois.edu   ✉ {cassano.f,a.guha}@northeastern.edu
⭘ https://github.com/bigcode-project/selfcodealign

## Abstract

Instruction tuning is a supervised fine-tuning approach that significantly improves the ability of large language models (LLMs) to follow human instructions. For programming tasks, most models are finetuned with costly human-annotated instruction-response pairs or those generated by large, proprietary LLMs, which may not be permitted. We propose SelfCodeAlign, the first fully transparent and permissive pipeline for self-aligning code LLMs without extensive human annotations or distillation. SelfCodeAlign employs the same base model for inference throughout the data generation process. It first extracts diverse coding concepts from high-quality seed snippets to generate new tasks. It then samples multiple responses per task, pairs each with test cases, and validates them in a sandbox environment. Finally, passing examples are selected for instruction tuning. In our primary experiments, we use SelfCodeAlign with CodeQwen1.5-7B to generate a dataset of 74k instruction-response pairs. Finetuning on this dataset leads to a model that achieves a 67.1 pass@1 on HumanEval+, surpassing CodeLlama-70B-Instruct despite being ten times smaller. Across all benchmarks, this finetuned model consistently outperforms the original version trained with OctoPack, the previous state-of-the-art method for instruction tuning without human annotations or distillation. Additionally, we show that SelfCodeAlign is effective across LLMs of various sizes, from 3B to 33B, and that the base models can benefit more from alignment with their own data distribution. We further validate each component's effectiveness in our pipeline, showing that SelfCodeAlign outperforms both direct distillation from GPT-4o and leading GPT-3.5-based distillation methods, such as OSS-Instruct and Evol-Instruct. SelfCodeAlign has also led to the creation of StarCoder2-Instruct, the first fully transparent, permissively licensed, and self-aligned code LLM that achieves state-of-the-art coding performance. Overall, SelfCodeAlign shows for the first time that a strong instruction-tuned code LLM can result from self-alignment rather than distillation.

## 1 Introduction

Recent studies have demonstrated the outstanding performance of large language models (LLMs) [33, 40, 19, 57, 45, 69, 8, 70] in various code-related tasks, *e.g.,* program synthesis [8, 3], program repair [78, 27, 24, 79, 73], code optimization [59, 9], code completion [11, 40, 19], code translation [56, 1, 51], software testing [32, 10, 42, 77], and software agents [80, 67, 75, 37]. The reason is that modern LLMs are pre-trained over trillions of code tokens in the wild using various training objectives (as such next-token prediction [52]), making the base models natively good at understanding and generating code snippets. Furthermore, to fully unleash the power of LLMs, the base models are

38th Conference on Neural Information Processing Systems (NeurIPS 2024).

typically further fine-tuned on high-quality instruction-following data to boost their performance in following natural language instructions and solving more general software engineering tasks [25]. This step is known as *instruction tuning* [50].

Curating high-quality data for instruction tuning is crucial yet challenging. One source of acquiring instruction data is to employ human annotation [50]. For example, Llama-3 [14] uses a corpus of 10 million human-annotated examples in instruction tuning. Due to the high cost of human annotation, knowledge distillation is widely adopted to train a weaker LLM with outputs generated by stronger LLMs [18]. However, distillation may violate the terms of service [48, 17, 2] of proprietary LLMs and the prerequisite of using a stronger LLM limits its generalizability. Therefore, recent proposals focus on instruction tuning without relying on human annotation or distillation [34, 60, 82]. One cornerstone work along this direction is SELF-INSTRUCT [68], which finetunes GPT-3 with self-generated instruction data using in-context learning.

There is a growing number of instruction-tuned open-source LLMs in the code domain. However, some models, such as DeepSeek-Coder [19], Llama-3 [14], and CodeQwen1.5 [64], either use propri- etary data or do not disclose their instruction-tuning strategies. Others, including WizardCoder [41], Magicoder [72], WaveCoder [81], and OpenCodeInterpreter [83], rely on knowledge distillation. The only exception is OctoCoder [43], which is instruction-tuned over heavily filtered GitHub commits, with commit messages as instructions and the changed code as responses, as well as data from Ope- nAssistant, a human-generated corpus of user-assistant conversations [29]. Despite its transparency and permissive licensing, OctoCoder's performance, at 32.9 HumanEval+ pass@1, lags behind other mainstream code LLMs. Meanwhile, previous attempts at applying SELF-INSTRUCT for code generation have resulted in performance degradation over training on natural instruction-response pairs [43]. Our findings imply that effective self-alignment requires a combination of data diversity control and response validation, which is not present in the traditional SELF-INSTRUCT approach.

In this paper, we propose SelfCodeAlign, the first fully transparent pipeline to successfully self-align base code LLMs with purely self-generated instruction data. First, SelfCodeAlign extracts diverse coding concepts from high-quality seed functions in The Stack V1 [28], a large corpus of permissively licensed code. Next, using these concepts, we prompt the base model to generate new coding tasks through in-context learning. We then instruct the base model to produce multiple responses for each task, each paired with test cases for self-validation. Finally, we select only the instruction-response pairs that pass the test cases. This method ensures the model practices various coding concepts and validates the consistency between instructions and responses.

To evaluate our method, we train CodeQwen1.5-7B, a state-of-the-art open-source base LLM for code, on both a dataset generated with SelfCodeAlign and OctoPack, a naturally-generated and meticulously-filtered dataset used for training OctoCoder [43]. We benchmark both, along with OctoCoder and other models, on a series of tasks: code generation (both function- and class- level) [38, 21, 76, 13], data science programming [30], and code editing [6]. On all tasks, training CodeQwen with SelfCodeAlign significantly improves performance over the base model and over training it on OctoPack. For instance, on HumanEval+, our model achieves a pass@1 score of 67.1, 21.4 points higher than CodeQwen1.5-7B and 16.5 points higher than CodeQwen1.5-7B-OctoPack. This highlights the effectiveness of our synthetic data generation method compared to natural data in enhancing the capabilities of code LLMs.

In the component analysis, we justify the different components of the pipeline. We demonstrate that SelfCodeAlign is general to different LLMs whose sizes range from 3B to 33B. In particular, we find that a base LLM could learn more effectively from data within its own distribution than a shifted distribution from a teacher LLM. Additionally, we show that seed selection, concept generation, and execution filtering all contribute positively to the pipeline. Furthermore, on HumanEval+, Self- CodeAlign (67.1 pass@1) outperforms state-of-the-art, GPT-3.5-Turbo-based distillation methods, including OSS-Instruct [72] (61.6) and Evol-Instruct [65] (59.1), as well as direct output distillation from GPT-4o [49] (65.9).

SelfCodeAlign has also led to the creation of StarCoder2-Instruct, the first fully transparent, permis- sively licensed, and self-aligned code LLM that achieves state-of-the-art coding performance. We discuss StarCoder2-Instruct in Appendix A.

Overall, we make the following main contributions: *(i)* We introduce SelfCodeAlign, the first fully transparent and permissive pipeline for self-aligning code LLMs to follow instructions. Our method

does not rely on extensive human annotations or distillation from larger models. *(ii)* We generate a series of datasets using SelfCodeAlign and train multiple models on these datasets, which will all be released to the public. *(iii)* We thoroughly evaluate our method on a multitude of tasks, showing strong performance across all the evaluated models. *(iv)* Our experiments demonstrate that training models on their own data can be more effective than using data from stronger, but distributionally different, teacher models when they don't have a huge performance gap. *(v)* Finally, we run a comprehensive component analysis that verifies the positive contribution of each component in SelfCodeAlign.

## 2 SelfCodeAlign: Self-Alignment for Code Generation

Figure 1 illustrates an overview of our SelfCodeAlign technique. It first generates diverse instructions by extracting coding concepts from high-quality seed snippets. This process resembles OSS-Instruct [72], which employs GPT-3.5-Turbo to convert random snippets into instructions. However, our method uses the base model exclusively and incorporates a separate concept generation phase that we prove beneficial in §4.3. SelfCodeAlign then generates several responses for each task, pairing each with test cases for sandbox execution, and finally chooses passing examples for instruction tuning. Example outputs from each step are listed in Appendix D.1. In the following sections, we provide detailed explanations of these steps.

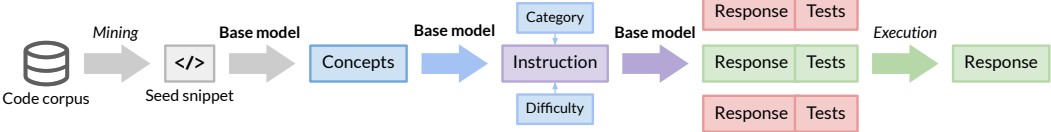

Figure 1: Overview of SelfCodeAlign.

### 2.1 Seed Snippets Collection

SelfCodeAlign starts by collecting a set of seed code snippets from The Stack V1. In this step, it's crucial to ensure that the seed snippets are diverse and high-quality, as they will be used as the starting point for generating instructions and responses. To collect the seed snippets, we extract all Python functions with docstrings from The Stack V1, and then apply a series of filtering rules to ensure the quality of the seed snippets. In total, we collect 250k Python functions from 5M functions with docstrings in The Stack V1, which were filtered by running the Pyright type checker, removing benchmark items, filtering out functions with poor documentation, and removing near-duplicates. Appendix B details this process in depth.

### 2.2 Diverse Instruction Generation

After collecting the seed functions, we perform Self-OSS-Instruct, our adaptation of OSS-Instruct [72] for self-alignment, to generate diverse instructions. In detail, we employ in-context learning to let the base model self-generate instructions from the given seed code snippets. This process utilizes 21 carefully designed few-shot examples listed in Appendix E. The instruction generation procedure is divided into the following two steps:

- **Concepts extraction:** For each seed function, we prompt the base model to produce a list of code concepts present within the function. Code concepts refer to the foundational principles and techniques used in programming, such as pattern matching and data type conversion.
- **Instruction generation:** We then prompt the base model to self-generate a coding task conditioned on the identified code concepts and two additional attributes, difficulty (easy/medium/hard) and category (function/class/program implementation), which we randomly sample to enrich the diversity of the generated instructions.

### 2.3 Response Generation and Self-Validation

Given the instructions generated from Self-OSS-Instruct, our next step is to match each instruction with a high-quality response. Prior practices commonly rely on distilling responses from stronger

teacher models, such as GPT-4, which hopefully exhibit higher quality. However, distilling proprietary models leads to non-permissive licensing and a stronger teacher model might not always be available. More importantly, teacher models can be wrong as well, and the distribution gap between teacher and student can be detrimental.

We propose to self-align the base model by explicitly instructing the model to generate tests for self-validation after it produces a response interleaved with natural language. This process is similar to how developers test their code implementations. Specifically, for each instruction, the base model samples multiple outputs of the format *(response, tests)*, and we filter out those responses falsified by the test execution under a sandbox environment. We then randomly select one passing response per instruction to the final instruction tuning dataset.

## 3   Main Evaluation

In this section, we comprehensively evaluate SelfCodeAlign over a diverse set of coding tasks:

- **Function generation (§3.1):** Given a natural-language description, LLMs are asked to generate a self-contained function whose correctness and efficiency is checked through test execution [8, 3, 38, 21, 76, 39].

- **Class generation (§3.2):** Given a code skeleton with both class- and method-level information, LLMs are asked to generate the class and its methods [13].

- **Data science programming (§3.3):** Given a description of a data science task and a partial code snippet, LLMs are asked to complete the code snippet to pass corresponding tests [30].

- **File-level code editing (§3.4):** Provided with the contents of a file, the model is asked to edit the program following a natural language instruction [6].

### 3.1   Function-level Code Generation

Table 1: Pass@1 (%) of different LLMs on EvalPlus computed using greedy decoding.

| Model | Instruction data | Benchmark | | Artifact | | |
| --- | --- | --- | --- | --- | --- | --- |
| | | HumanEval+ | MBPP+ | Transparent | Non-proprietary | Non-distilled |
| GPT-4-Turbo [47] | Not disclosed | 81.7 | 70.7 | ○ | ○ | ● |
| Mistral Large [22] | Not disclosed | 62.8 | 56.6 | ○ | ○ | ● |
| Gemini Pro [63] | Proprietary | 55.5 | 57.9 | ○ | ○ | ● |
| Llama3-70B-Instruct [14] | Proprietary | 70.7 | 66.4 | ○ | ○ | ● |
| CodeLlama-70B-Instruct [57] | Proprietary | 65.2 | 61.7 | ○ | ○ | ● |
| WizardCoder-33B-v1.1 [41] | GPT distillation | 73.2 | 66.9 | ○ | ● | ○ |
| OpenCodeInterpreter-DS-33B [83] | GPT distillation | 73.8 | 67.7 | ● | ● | ○ |
| Magicoder-S-DS-6.7B [72] | GPT distillation | 70.7 | 65.4 | ● | ● | ○ |
| DeepSeekCoder-33B-Instruct [19] | Not disclosed | 75.0 | 66.7 | ○ | - | - |
| CodeQwen1.5-7B-Chat [64] | Not disclosed | 77.7 | 67.2 | ○ | - | - |
| Snowflake Arctic (480B) [55] | Not disclosed | 64.3 | 64.3 | ○ | - | - |
| Mixtral-8x22B-Instruct-v0.1 [23] | Not disclosed | 70.1 | 62.9 | ○ | - | - |
| Command-R+ (104B) [16] | Not disclosed | 56.7 | 58.6 | ○ | - | - |
| Mixtral-8x7B-Instruct-v0.1 [23] | Not disclosed | 39.6 | 49.7 | ○ | - | - |
| OctoCoder-16B [43] | Publicly available | 32.9 | 49.1 | ● | ● | ● |
| StarCoder2-15B [40] | - | 37.8 | 53.1 | ● | ● | ● |
| CodeQwen1.5-7B-Base [64] | - | 45.7 | 60.2 | ○ | - | - |
| CodeQwen1.5-7B-OctoPack | Publicly available | 50.6 | 63.2 | ● | ● | ● |
| SelfCodeAlign-CQ-7B | Self-generated | 67.1 | 65.2 | ● | ● | ● |

**HumanEval+ and MBPP+.** HumanEval [8] and MBPP [3] are the two most widely-used benchmarks for function-level code generation. We use their test augmented versions, *i.e.,* HumanEval+ and MBPP+, with 80×/35× more test cases for rigorous evaluation [38].

As baselines, we consider a diverse set of state-of-the-art instruction-tuned models over various dimensions, including weight openness, data openness, transparency, and performance. Table 1 compares the pass@1 of the self-aligned SelfCodeAlign-CQ-7B against other baseline models on

HumanEval+ and MBPP+. Among those trained using a fully transparent pipeline without any proprietary data or distillation, SelfCodeAlign-CQ-7B stands out as the best LLM by drastically outperforming the base model, OctoCoder-16B, StarCoder2-15B, and CodeQwen1.5-7B-OctoPack. Meanwhile, compared to much larger models, SelfCodeAlign-CQ-7B outperforms Arctic, Command-R+, and Mixtral-8x7B-Instruct, while closely matching Mixtral-8x22B-instruct. Even compared to LLMs trained using proprietary data (*e.g.,* manually annotated), SelfCodeAlign-CQ-7B remains competitive, surpassing Gemini Pro, Mistral Large, and CodeLlama-70B-Instruct. Additionally, SelfCodeAlign-CQ-7B, fine-tuned on purely self-generated data, closely rivals models finetuned with distillation-based or non-transparent synthetic data.

**LiveCodeBench.** In subsequent evaluations, we benchmark our model against state-of-the-art open-source LLMs of similar sizes for a fair comparison. LiveCodeBench [21] is a benchmark for contamination-free evaluation. It features 400 recent Python algorithm challenges from May 2023 to February 2024. These tasks are curated from online judge websites such as Codeforce and LeetCode, each with over 20 test cases on average. While LiveCodeBench is a holistic benchmark covering four problem types, we focus on the code generation task for assessing LLM function generation.

Table 2 reports the pass@1 results for problem subsets created after three specific start dates. It shows that SelfCodeAlign-CQ-7B consistently outperforms most baseline models and closely matches CodeQwen1.5-7B-Chat. In addition, moving the start date forward has minimal impact on the pass@1 of SelfCodeAlign-CQ-7B, indicating that our pipeline is less likely to suffer from contamination.

Table 2: Pass@1 (%) of LLMs on LiveCodeBench. Newer start dates imply lower contamination risk.

| Model | Start date | | |
|---|---|---|---|
| | 2023-09-01 | 2023-07-01 | 2023-05-01 |
| DeepSeek-Coder-6.7B-Instruct | 19.2 | 20.8 | 21.6 |
| CodeGemma-7B-IT | 15.2 | 14.1 | 13.6 |
| Llama-3-8B-Instruct | 18.3 | 18.4 | 17.3 |
| CodeQwen1.5-7B-Base | 19.3 | 20.7 | 21.8 |
| CodeQwen1.5-7B-Chat | **23.2** | **24.1** | **25.0** |
| OctoCoder-16B | 12.6 | 11.2 | 9.8 |
| StarCoder2-15B | 14.5 | 14.7 | 15.4 |
| CodeQwen1.5-7B-OctoPack | 19.3 | 21.8 | 22.5 |
| SelfCodeAlign-CQ-7B | **22.4** | **22.8** | **23.4** |

**EvoEval.** To mitigate the impact of potential data contamination, EvoEval [76] includes 828 programming problems created by prompting GPT-4 to evolve original HumanEval tasks across 5 semantic-altering and 2 semantic-preserving benchmarks. Following the leaderboard of EvoEval, we use the 5 semantic-altering benchmarks, each of which has 100 problems.

Table 3 shows that SelfCodeAlign-CQ-7B achieves the best pass@1 score among all transparently finetuned models. Meanwhile, it also surpasses most open LLMs (except CodeQwen1.5-7B-Chat) trained on unknown instruction-tuning data.

Table 3: Pass@1 (%) of code LLMs on EvoEval.

| Model | Average | Difficult | Creative | Subtle | Combine | Tool use |
|---|---|---|---|---|---|---|
| DeepSeek-Coder-6.7B-Instruct | 41.4 | 40 | 37 | 61 | 18 | 51 |
| CodeGemma-7B-IT | 35.4 | 31 | 32 | 49 | 9 | 56 |
| Llama-3-8B-Instruct | 40.6 | 34 | 39 | 57 | 15 | 58 |
| CodeQwen1.5-7B-Base | 36.2 | 26 | 30 | 46 | 18 | 61 |
| CodeQwen1.5-7B-Chat | **48.0** | 39 | 38 | 71 | 31 | 61 |
| OctoCoder-16B | 30.6 | 19 | 26 | 43 | 11 | 54 |
| StarCoder2-15B | 25.8 | 16 | 19 | 41 | 5 | 48 |
| CodeQwen1.5-7B-OctoPack | 42.2 | 35 | 36 | 59 | 22 | 59 |
| SelfCodeAlign-CQ-7B | **43.6** | 33 | 40 | 60 | 20 | 65 |

**EvalPerf.** While the earlier benchmarks focus on code correctness, we use EvalPerf [39] to evaluate the efficiency of LLM-generated code. EvalPerf includes 118 performance-exercising tasks with computation-intensive test inputs to fully exercise the efficiency of LLM-generated code.

Since code efficiency only matters when the generated code is correct, in Table 4 we only evaluate baselines that can achieve a decent pass@1 (*i.e.,* over 50%) on HumanEval+. Specifically, we run EvalPerf by following its default settings: *(i)* Each model generates 100 samples per task at the temperature of 1.0; *(ii)* We evaluate the efficiency of up to 20 correct samples per model for tasks where it can at least generate 10 passing samples; and *(iii)* Finally we rank the models based on their win rates, where each model pair compares their differential performance score (DPS) over the common set of passing tasks. Notably, DPS is a LeetCode-inspired metric that indicates the overall efficiency ranking of submissions. For example, Table 4 shows that SelfCodeAlign-CQ-7B achieves a DPS of 79.9, indicating that its correctly generated solutions can overall outperform or match the efficiency 79.9% of reference solutions across various efficiency levels.

Table 4 shows that SelfCodeAlign-CQ-7B ranks second among the evaluated models of comparable size. Specifically, SelfCodeAlign-CQ-7B is only next to DeepSeek-Coder-6.7B-Instruct whose training data is not disclosed. Surprisingly, the efficiency of SelfCodeAlign-CQ-7B-generated code surpasses many recent open models trained using private data, including the latest Llama-3.1-8B-Instruct.

Table 4: Ranking of model code efficiency based on the EvalPerf win rates, which are computed over the common set of passing tasks for each model pair. Each model generates 100 samples per task at a temperature 1.0. To exemplify differential performance score (DPS) with SelfCodeAlign-CQ-7B, it means its generations if correct can match the efficiency of 79.9% LLM samples.

| Model | DPS (%) | pass@1 (%) | Win-rate (%) |
|---|---|---|---|
| DeepSeek-Coder-6.7B-Instruct | 83.6 | 73.6 | 63.9 |
| Llama-3.1-8B-Instruct | 80.9 | 64.3 | 52.1 |
| Llama-3-8B-Instruct | 77.0 | 43.7 | 51.5 |
| CodeQwen1.5-7B-Chat | 80.7 | 74.1 | 51.2 |
| CodeQwen1.5-7B-OctoPack | 74.0 | 49.1 | 26.9 |
| SelfCodeAlign-CQ-7B | 79.9 | 65.2 | 54.0 |

## 3.2 Class-level Code Generation

We evaluate code LLMs on class-level code generation using ClassEval [13], a collection of 100 class-level Python code generation tasks, covering 100 classes and 410 methods, with an average of 33 tests per class and 8 tests per method.

Following the ClassEval paper [13], we set the maximum model context size as 2048 tokens and report the best class-level pass@1 (and corresponding method-level pass@1) of each model among three generation strategies: (i) *Holistic Generation*: generating the entire class given a class skeleton, (ii) *Incremental Generation*: generating class methods iteratively by putting earlier generated methods in the prompt, and (iii) *Compositional Generation*: generating each class method independently without looking at other methods. Specifically, class-level pass@1 in Table 5 refers to the pass rate of generated classes given *both* the method- and class-level tests. In contrast, method-level pass@1 is computed by *only* checking if the generated methods can pass the method-level tests. Table 5 shows, in terms of class-level performance, SelfCodeAlign-CQ-7B is the best transparently finetuned model, surpassing the second-best transparent model (*i.e.,* CodeQwen1.5-7B-OctoPack) by 28%, while performing no worse than those using unknown or proprietary instruction-tuning data. For method generation, SelfCodeAlign-CQ-7B also stands out in transparently finetuned models.

## 3.3 Data Science Programming

DS-1000 [30] is a benchmark of 1000 realistic data science challenges across 7 popular Python data science libraries. In DS-1000, a model must complete a partial code snippet to solve the problem. The solution is then evaluated through test execution. Table 6 shows that SelfCodeAlign-CQ-7B, despite being trained on limited data science code, stands out as the best in the transparent model category, while remaining competitive among the other evaluated baselines.

Table 5: Pass@1 (%) of code LLMs on ClassEval using greedy decoding.

| Model | Class-level | Method-level |
|---|---|---|
| DeepSeek-Coder-6.7B-Instruct | **27.0** | **57.2** |
| CodeGemma-7B-IT | 21.0 | 44.8 |
| Llama-3-8B-Instruct | 23.0 | 52.4 |
| CodeQwen1.5-7B-Base | 23.0 | 52.8 |
| CodeQwen1.5-7B-Chat | **27.0** | 54.6 |
| OctoCoder-16B | 19.0 | 38.0 |
| StarCoder2-15B | 9.0 | 24.9 |
| CodeQwen1.5-7B-OctoPack | 21.0 | 45.2 |
| SelfCodeAlign-CQ-7B | **27.0** | **52.6** |

Table 6: Pass@1 (%) on DS-1000 with temperature 0.2 and top-p 0.95 over 40 samples, following the same hyperparameter setting used in StarCoder2 [40].

| Model | Avg. | Pandas | NumPy | Matplotlib | TensorFlow | SciPy | Sklearn | PyTorch |
|---|---|---|---|---|---|---|---|---|
| DeepSeek-Coder-6.7B-Instruct | 44.6 | 34.0 | 51.1 | 58.4 | 45.9 | 34.2 | 45.8 | 50.6 |
| CodeGemma-7B-IT | 30.8 | 21.9 | 34.4 | 54.7 | 25.1 | 21.8 | 22.6 | 34.5 |
| Llama-3-8B-Instruct | 31.1 | 21.5 | 33.1 | 51.9 | 34.4 | 25.2 | 23.8 | 37.2 |
| CodeQwen1.5-7B-Base | 32.4 | 21.6 | 35.9 | 56.7 | 28.8 | 28.2 | 30.9 | 23.8 |
| CodeQwen1.5-7B-Chat | **47.1** | 34.4 | 51.7 | 67.2 | 46.0 | 38.9 | 47.9 | 52.8 |
| OctoCoder-16B | 28.3 | 13.1 | 34.0 | 53.8 | 22.4 | 22.8 | 30.0 | 25.9 |
| StarCoder2-15B | 38.9 | 26.2 | 45.8 | 61.4 | 38.1 | 36.0 | 40.5 | 22.5 |
| CodeQwen1.5-7B-OctoPack | 38.2 | 26.7 | 42.6 | 61.8 | 37.7 | 32.7 | 36.6 | 31.4 |
| SelfCodeAlign-CQ-7B | **39.1** | 28.2 | 42.6 | 57.2 | 38.3 | 35.6 | 42.8 | 33.3 |

## 3.4 Code Editing

We further evaluate LLMs on code editing tasks using the CanItEdit benchmark [6], comprised of 210 code editing tasks from three change kinds (70 tasks each): corrective (fixing bugs), adaptive (adding new features), and perfective (improving existing features). The tasks are evaluated based on the correctness of the generated code changes, according to a set of hidden test cases. For each task, the model is given as input the original code snippet and a natural-language instruction describing the desired code change; then it is expected to produce an updated code snippet that satisfies the instruction. We follow the setting from the original benchmark [6] to generate 20 completions per task at a temperature of 0.2. Table 7 reports the pass@1 for each change kind and the average pass@1 across all tasks. Despite not being specifically tuned for code editing, SelfCodeAlign-CQ-7B exhibits strong performance on CanItEdit, achieving a pass@1 of 39.0%, outperforming all other models except CodeQwen1.5-Chat, whose instruction tuning details are not disclosed.

Table 7: Pass@1 (%) of code LLMs on CanItEdit.

| Model | Average | Corrective | Adaptive | Perfective |
|---|---|---|---|---|
| DeepSeek-Coder-6.7B-Instruct | 36.3 | 34.9 | 38.8 | 35.3 |
| CodeGemma-7B-IT | 34.2 | 30.9 | 39.3 | 32.5 |
| Llama-3-8B-Instruct | 36.0 | 34.9 | 39.1 | 34.0 |
| CodeQwen1.5-7B-Base | 38.4 | 34.7 | 45.6 | 34.9 |
| CodeQwen1.5-7B-Chat | **39.9** | 38.1 | 46.6 | 35.1 |
| OctoCoder-16B | 30.2 | 38.4 | 31.6 | 20.5 |
| StarCoder2-15B | 36.7 | 32.1 | 43.8 | 34.2 |
| CodeQwen1.5-7B-OctoPack | 36.5 | 36.9 | 40.6 | 31.9 |
| SelfCodeAlign-CQ-7B | **39.0** | 37.4 | 42.4 | 37.2 |

# 4 Component Analysis

In this section, we extensively study how different components contribute to the SelfCodeAlign pipeline. To make the comparison tractable, we fix a subset of seed code snippets by randomly sampling 37k examples from the 250k corpus and evaluate finetuned models on HumanEval+ [38].

## 4.1 Self-Alignment with Different Models

To assess whether SelfCodeAlign is generalizable and how performance varies with finetuning data generated by different models, we run the same data generation pipeline end to end with different LLMs. We include four diverse state-of-the-art model architectures and sizes ranging from 3B to 33B to observe how SelfCodeAlign performs across small, medium, and large-scale LLMs.

Table 8 shows the comparison and guides us to reach the following findings. Looking at the diagonal cells, SelfCodeAlign consistently improves the performance of the base models with varying sizes, from 3B to 33B. Comparing each diagonal cell and the cell immediately to its right (i.e., using base models with slightly better HumanEval+ performance as the teacher models), we can see that a base model may benefit more from self-generated data than a stronger teacher model, when they don't have a large performance gap. However, when the teacher model is clearly stronger, the base model learns better by distilling the teacher's knowledge. For example, StarCoder2-3B achieves higher pass@1 trained on its own data (35.4) compared to Llama-3-8B data (34.1), but when tuned with stronger models, StarCoder2-3B further improves (*e.g.,* 42.1 with DeepSeek-Coder-33B data). Also, the last row shows that a stronger model can still learn from a weaker model, but less effectively. We provide qualitative examples in Appendix D.2.

Table 8: HumanEval+ pass@1 when finetuning the base models on different data (37k seeds).

| Base model (pass@1) | Data-generation model | | | | |
|---|---|---|---|---|---|
| | StarCoder2-3B | Llama-3-8B | StarCoder2-15B | DeepSeek-Coder-33B | CodeQwen1.5-7B |
| StarCoder2-3B  (27.4) | 35.4 | 34.1 | 39.0 | **42.1** | 40.2 |
| Llama-3-8B  (29.3) | - | 42.7 | 40.2 | 41.5 | **43.3** |
| StarCoder2-15B  (37.8) | - | - | 55.5 | 53.0 | **57.3** |
| DeepSeek-Coder-33B  (44.5) | - | - | - | **65.9** | 62.2 |
| CodeQwen1.5-7B  (45.7) | 48.8 | 54.9 | 56.1 | 59.1 | **65.2** |

## 4.2 Effectiveness of Execution-based Filtering

The SelfCodeAlign pipeline samples multiple responses for an instruction and each response is equipped with self-generated test cases. Responses with failing tests are filtered out and each instruction will be paired with a randomly selected passing response. To answer the question of "to what extent is execution information helpful", in Table 9, we conduct 4 controlled experiments by varying how responses are selected while keeping the other components unchanged:

- **Random selection (all)**: pair each instruction with a random response without response filtering.
- **Random selection (subset)**: 15.6k subset of "Random selection (all)" for a consistent data amount.
- **Failures only**: pair each instruction with a failing response.
- **Passes only**: pair each instruction with a passing response.

Table 9: Pass@1 on HumanEval+ with different response selection strategies.

| Selection strategy | Data size | Execution pass rate | Pass@1 |
|---|---|---|---|
| Random selection (all) | 27.7k | 24.1% | 61.6 |
| Random selection (subset) | 15.6k | 24.2% | 61.6 |
| Failures only | 15.6k | 0% | 57.9 |
| Passes only | 15.6k | 100.0% | **65.2** |

First, we can observe that random pairing performs worse than using only passing examples, both when data sizes are aligned and when they scale up by 1.8×. Meanwhile, the "Failure only" setting

results in the worst performance where we deliberately use failing responses for each instruction. These results suggest the importance of execution filtering and code correctness for self-alignment.

## 4.3 Importance of Seed Selection and Concepts Generation

For instruction generation, SelfCodeAlign applies Self-OSS-Instruct that first selects high-quality seed code snippets, then mines code concepts from the seeds, and finally generates the instructions. To validate the usefulness of concept generation and high-quality seeds, we compare two variants of SelfCodeAlign in Table 10: 1) directly generating instructions from seeds, where the model produces an instruction based solely on a seed snippet, and 2) using the default pipeline except for the initial seeds, where random snippets are sampled from different code documents in The Stack V1.

Table 10: Pass@1 on HumanEval+ using different seeds and pipelines.

| Source of seeds | Pipeline | Pass@1 |
|---|---|---|
| Filtered functions | Seed → instruction | 59.8 |
| Random snippets | Seed → concepts → instruction | 64.0 |
| Filtered functions | Seed → concepts → instruction | **65.2** |

It is shown that directly generating instructions from seeds leads to the poorest performance. This is because a direct generation from seeds requires the seed snippet to be presented in the context, whose format is not well represented in the wild and may not be in distribution for the model. The generated instructions will then be distracted and thus be of lower quality. Concept generation neutralizes this effect and produces more realistic and natural instructions. Using random snippets produces a more diverse but less coherent set of concepts, leading to slightly worse performance compared to using high-quality seeds. Appendices D.3 and D.4 illustrate some qualitative examples.

## 4.4 Comparing Self-Alignment to Distillation

Table 11: SelfCodeAlign versus distillation using CodeQwen1.5-7B as the base model.

| Method | Dataset size | Teacher model | Execution filtering | Pass@1 |
|---|---|---|---|---|
| Evol-Instruct | 74k | GPT-3.5-Turbo | ○ | 59.1 |
| OSS-Instruct | 74k | GPT-3.5-Turbo | ○ | 61.6 |
| Direct distillation | 74k | GPT-4o | ○ | 65.9 |
| SelfCodeAlign | 74k | CodeQwen1.5-7B | ● | **67.1** |

To compare self-alignment with distillation, we evaluate SelfCodeAlign against two state-of-the-art distillation methods for code instruction tuning: OSS-Instruct [72] and Code Evol-Instruct [65]. We use the official OSS-Instruct dataset. As the official implementation of Code Evol-Instruct is unavailable, we opt for the most popular open-source version [44] on Hugging Face. Both datasets are generated using GPT-3.5-Turbo [46] and we randomly select their subsets to match the 74k samples generated by SelfCodeAlign. Table 11 shows that SelfCodeAlign substantially outperforms both methods, indicating the strength and promising future of self-alignment for code. Additionally, SelfCodeAlign outperforms direct distillation, where we use the same set of SelfCodeAlign instructions but rely on GPT-4o [49] to generate each response at temperature 0. This suggests that weaker models, combined with more post-validation compute, can produce higher-quality responses.

## 5 Related Work

**Instruction tuning for code.** To build more powerful code assistants, pre-trained code models are fine-tuned over a small amount of high-quality instruction-response pairs that are either collected from real-world [43] or synthetically generated [7, 57, 41, 72]. This step is known as instruction tuning. OctoPack [43] compiles a large set of real-world Git commits which are partially used for code fine-tuning. Code Alpaca [7] applies SELF-INSTRUCT to the code domain, which prompts ChatGPT to generate synthetic instruction data for code. Similarly, the instruction data for CODELLAMA [57]

includes coding problems generated by prompting LLAMA 2 [66] and solutions and tests by prompting base CODELLAMA. Code Evol-Instruct [41] uses harder programming challenges as instruction data to fine-tune more capable models. Specifically, Code Evol-Instruct prompts ChatGPT with heuristics to evolve existing instruction data to more challenging and complex ones. Besides data complexity, the widely-adopted [14, 62, 71] OSS-INSTRUCT [72] looks at the data *diversity* and *quality* dimension. Specifically, given a source code snippet, OSS-INSTRUCT prompts ChatGPT to get inspired and imagine potential instruction-response pairs, which inherit the diversity and quality of sampled code snippets. Besides instruction tuning, recent work on training code LLMs for performance improvement also explores multi-turn code generation [83], model merging [12], preference tuning [74, 36], and reinforcement learning [15]. Recently, various strong instruction-tuned code models have been released by major organizations [19, 64]. However, their instruction-tuning recipes (*e.g.,* data and strategies) are not fully disclosed. This lack of transparency underscores the need for fully transparent and permissive instruction-tuning methods to advance the field.

**Self-alignment.** Self-alignment is an approach to instruction tuning that utilizes an LLM to learn from its own output without depending on an existing well-aligned teacher LLM. SELF-INSTRUCT [68] is one of the first endeavors that allow GPT-3 to improve itself by generating new instructions and responses for instruction-tuning using its in-context learning capability. SELF-ALIGN [61], based on in-context learning, utilizes topic-guided SELF-INSTRUCT for instruction generation and pre-defines principles to steer the LLM towards desired responses. These instruction-response pairs are used to fine-tune the base model, followed by a final refinement stage to ensure the model produces in-depth and detailed responses. Instruction backtranslation [35] offers an alternative self-alignment method by initially training a backward model to generate instructions from unlabeled web documents using limited seed data. It then iteratively produces new instructions from new web documents and selects high-quality data for self-training. Most code LLM work targets knowledge distillation. Haluptzok et al. [20] share a relevant idea to our work but only consider program puzzles specified in symbolic forms. This setting cannot be generalized to real-world tasks with natural language involved.

# 6  Limitations and Future Work

We limit our data generation within a ∼3000 window, skewing our distribution towards medium-sized samples. Therefore, generating and training on long-context instruction-response pairs can be a promising avenue [4]. Second, we gather several negative samples during response generation, which are currently filtered out. These negatives could be used in a reinforcement-learning loop to steer the model away from incorrect responses [31, 53]. Furthermore, the good responses are labeled by test execution, while the generated unit tests might be erroneous, calling for research to study and improve the generation of valid test cases. Finally, we plan to apply SelfCodeAlign to more challenging domains such as complex program generation [84] and agentic software engineering [26].

# 7  Conclusion

We introduce SelfCodeAlign, the first fully transparent and permissive pipeline for self-aligning code LLMs without extensive human annotations or distillation. SelfCodeAlign-CQ-7B, finetuned from CodeQwen1.5-7B using SelfCodeAlign, outperforms the $10\times$ larger CodeLlama-70B-Instruct on HumanEval+ and consistently surpasses CodeQwen1.5 trained with OctoPack on all studied benchmarks. We evaluate SelfCodeAlign across various model sizes, illustrating that stronger base models benefit more from self-alignment than distillation. We also examine the effectiveness of different components in the pipeline, showing that SelfCodeAlign is better than GPT-3.5 and GPT-4o distillation. Overall, we demonstrate for the first time that a strong instruction-tuned code LLM can be created through self-alignment, without expensive human annotations or distillation.

# Acknowledgements

We thank all the reviewers for their insightful comments and suggestions for our paper. This work was partially supported by NSF grants CCF-2131943, SES-2326173, and Kwai Inc, as well as API credits from the OpenAI Researcher Access Program.

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

## A  StarCoder2-Instruct: Fully Transparent and Permissive Self-Alignment for Code Generation

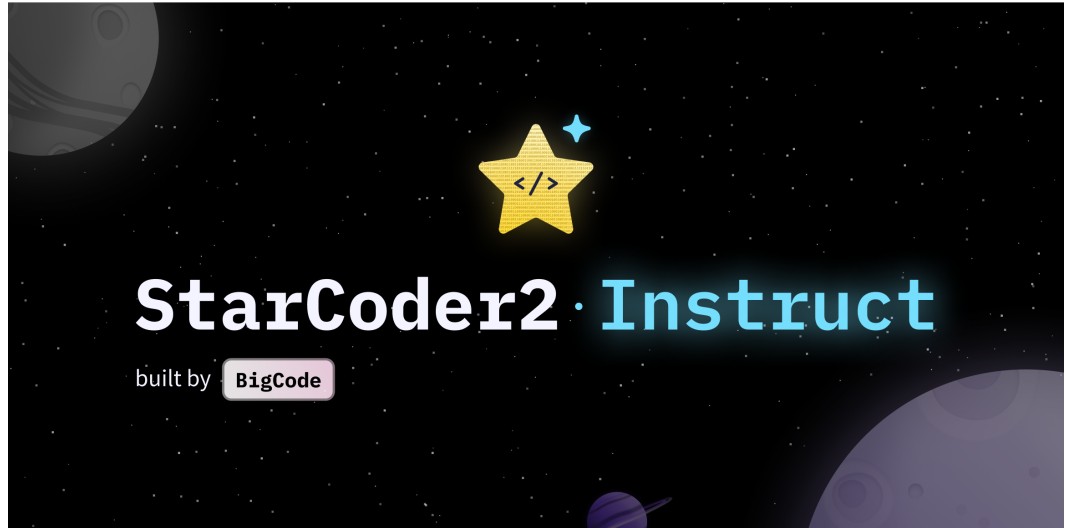

StarCoder2-Instruct is the very first entirely self-aligned code LLM created with an earlier version of SelfCodeAlign. The pipeline uses StarCoder2-15B to generate thousands of instruction-response pairs, which are then used to finetune StarCoder2-15B itself without any human annotations or distilled data from huge and proprietary LLMs. StarCoder2-15B-Instruct achieves a 72.6 HumanEval score, surpassing the 72.0 score of CodeLlama-70B-Instruct. More details are explained in the blog: https://huggingface.co/blog/sc2-instruct.

## B  Seed Data Curation

```python
import torch

def one_hot(y, num_dim=10):
    """
    One Hot Encoding, similar to 'torch.eye(num_dim).index_select(dim
    =0, index=y)'
    :param y: N-dim tenser
    :param num_dim: do one-hot labeling from '0' to 'num_dim-1'
    :return: shape = (batch_size, num_dim)
    """
    one_hot_y = torch.zeros(y.size(0), num_dim)
    if y.is_cuda:
        one_hot_y = one_hot_y.cuda()
    return one_hot_y.scatter_(1, y.view(-1, 1), 1.)
```

Listing 1: An example Python seed function from our final dataset.

In this section we describe in detail our seed gathering process, which is used to collect the initial set of seed Python functions for the concept extraction phase.

We provide an example seed function in Listing 1. All seeds in our dataset take on the format of: imports (if any), signature, docstring, implementation.

### B.1  Python Function Extraction

```
(
    (function_definition
      name: (identifier)
      body: (block .
```

```
        (expression_statement
            (string
                (string_start) @docstring.start
                (string_content)
                (string_end) @docstring.end)))) @function.def
    (#eq? @docstring.start "\\\"\\\"\\\"")
    (#eq? @docstring.end "\\\"\\\"\\\"")
)
```

Listing 2: The Tree-sitter query utilized for extracting Python functions with docstrings.

Our seed gathering process starts off by extracting all Python functions with docstrings from The Stack v1, which is a large dataset of code from GitHub. We accomplish this by utilizing the Tree-sitter parser and the query language it provides. Listing 2 provides the query utilized for matching each function, assuring that the functions live at the top-level of the program, and that they indeed contain a docstring. Utilizing this query, we extracted a total of 5,359,051 Python functions with docstrings. Interestingly, we found that roughly only 20% of Python functions contain any docstring.

## B.2 Quality Filtering and Transformations

After gathering our 5M Python functions, we apply a series of filtering and transformations steps. After all of our filtering rules, we are left with a dataset of 248,934 high-quality Python functions. These steps are a generalization of the dataset building pipeline in MultiPL-T [5], which only manages to produce half as many functions (those without imports). We detail each step of this process below.

**Import prediction** By naively extracting the functions from Python files, we may have lost import statements of external libraries that are utilized inside the function. To remedy this, we utilize the autoimport library to infer potential import statements for unbound identifiers in the function.

**Removing benchmark data** To enable a fair evaluation of our method, we decontaminate our seed dataset from examples that resemble prompts and solutions of items in the benchmarks on which we evaluate on. We accomplish this by checking if either the substring of the solution or prompt to each benchmark item exists in any function in the dataset.

**Return filtering** To aid in our self-validation step, we aim to include only functions that return a value, such that potential responses will contain test cases with complex expected values. We utilize Tree-sitter to filter any function that does not contain at least one return statement with an argument value.

**Type-checking** To further ensure the quality of our Python functions, we apply Pyright, a heuristic type-checker for Python, on all of our functions, and keep only ones passing the check. This step also ensures that no unbound identifiers are referenced inside the function.

**Docstring quality filtering** We find that several Python functions, while having defined a docstring, contain poor or misleading documentation. In aims of removing such functions, we employ StarCoder2-15B with a simple binary classification prompt, tasking the model to detect functions with poor docstrings. We remove all functions that were deemed poor quality by this classifier.

**Near-deduplication** As a final step, we wish to increase the diversity of seeds in our dataset. To accomplish this, we utilize MinHash with Locality-Sensitive Hashing and a Jaccard Similarity threshold of 0.5 to identify duplicate groups of functions in our seed dataset. We then only pick a single function from each group, and add it to our final dataset. We note that this is the same process utilized to deduplicate the pre-training dataset of StarCoder and StarCoder2 [33, 40].

## C Implementation Details

### C.1 Data Generation

We implement 21 few-shot examples of the form *(seed, property, instruction, response, tests)*, where coding concepts are encoded in the property of each example. Besides coding concepts and

programming language, a property includes a task category and a difficulty level that are randomly sampled during data generation. We use eight-shot for concept and instruction generation, and one-shot for response generation. During response generation, we explicitly guide the model to generate tests by concatenating the response and tests in the one-shot example. For the main experiment, if the test case follows a specified format, we additionally include it in the instruction body with a fifty percent chance to boost diversity. Table 12 shows the estimated cost for end-to-end data generation with different models.

Throughout the data generation pipeline, we follow [61] and choose a temperature at 0.7 to strike a balance between diversity and quality.

Table 12: End-to-end data generation time cost on 4×A100

| Base model | Seed data | Produced data | Generation cost | Finetuning cost |
|---|---|---|---|---|
| Llama-3-8B | 37k | 11k | 7h | 20min |
| CodeQwen1.5-7B | 37k | 16k | 7h | 30min |
| StarCoder2-15B | 37k | 15k | 12h | 2.5h |
| StarCoder2-3B | 37k | 12k | 6h | 16min |
| DeepSeek-Coder-33B | 37k | 15k | 83h | 3.5h |

## C.2 Execution

We implement a Docker-based execution server for code execution. This sandbox environment includes widely-used Python packages such as Numpy, PyTorch, and Pandas, allowing us to safely execute arbitrary Python code. Additionally, the server supports parallel requests to speed up validation.

## C.3 Training

Our overall hyperparameter choices are derived from existing good practices [41, 35, 72, 83]. We set the initial learning rate at 1e-5 for training on self-generated data and 2e-5 for training on data generated from other models. Empirically, we find this to be the optimal setting for both cases. We adopt a 0.05 warmup ratio and a linear scheduler. We use Adafactor [58] as our optimizer and choose a batch size of 64 with a sequence truncation length of 1280.

## C.4 Computater Resources

We primarily conduct data generation, training, and evaluation on a node equipped with 4 NVIDIA A100 PCI-E GPUs, 128 cores, and 512 GB of memory. For experiments involving DeepSeek-Coder, we use a node with 8 NVIDIA H100 GPUs. For DeepSeek-Coder, we utilize DeepSpeed ZeRO-3 [54] for training. For StarCoder2-15B, we use one A100 for training since otherwise it cannot fit the GPU memory due to the extra overhead caused by inter-GPU communication. For all the other experiments, we do a 4-GPU training using PyTorch's Distributed Data Parallel (DDP) module.

## C.5 License

As indicated in Table 13, each dataset's license inherits the license of the model that produces it. Additionally, each model's license combines both the base model's license and the data generation model's license. Our source code is licensed under Apache-2.0.

# D  Examples

## D.1  Seeds, Concepts, Instructions, Responses, and Tests

```
[Seed]
def get_email_domain(email):
    """Generates email domain from given email address."""
    return "www."+email.split("@")[-1]
```

Table 13: License of models and datasets

| Asset | Type | License |
|-------|------|---------|
| StarCoder2-3B | Model | BigCode OpenRAIL-M v1 |
| Llama-3-8B | Model | llama3 |
| CodeQwen1.5-7B | Model | tongyi-qianwen-research |
| StarCoder2-15B | Model | BigCode OpenRAIL-M v1 |
| DeepSeek-Coder-33B | Model | deepseek-license |
| ${DATASET} generated by ${MODEL} | Dataset | License of ${MODEL} |
| ${MODEL_A} finetuned on ${DATASET} generated by ${MODEL_B} | Model | License of ${MODEL_A} + License of ${MODEL_B} |
| Source code | Code | Apache-2.0 |

```
[Concepts]
['domain extraction from email addresses', 'URL manipulation', 'string
    manipulation']

[Instruction]
Use Python to create a class called `EmailProcessor`. The class should
    have methods to extract the domain name from an email address,
    split a URL into its components (protocol, host, port, path), and
    perform a simple regex search on a string. The class should also
    have a method to check if a URL is valid and can be opened in a
    web browser.

[Response]
Here's how you can create the `EmailProcessor` class:

```python
import re
from urllib.parse import urlparse

class EmailProcessor:

    @staticmethod
    def extract_domain(email: str) -> str:
        return email.split('@')[1]

    @staticmethod
    def split_url(url: str) -> dict:
        parsed_url = urlparse(url)
        return {
            'protocol': parsed_url.scheme,
            'host': parsed_url.hostname,
            'port': parsed_url.port,
            'path': parsed_url.path,
        }

    @staticmethod
    def regex_search(text: str, pattern: str) -> bool:
        return re.search(pattern, text) is not None

    @staticmethod
    def is_valid_url(url: str) -> bool:
        try:
            result = urlparse(url)
            return all([result.scheme, result.netloc])
        except ValueError:
```

```
            return False

    @staticmethod
    def open_url(url: str):
        import webbrowser
        webbrowser.open(url)
```

The `extract_domain` method extracts the domain name from an email
    address by splitting on the '@' symbol. The `split_url` method
    uses `urlparse` to split a URL into its components and returns
    them as a dictionary. The `regex_search` method uses `re.search`
    to perform a regex search on a string and returns True if a match
    is found, otherwise False. The `is_valid_url` method uses `
    urlparse` to validate a URL and returns True if it's valid,
    otherwise False. The `open_url` method uses `webbrowser.open` to
    open a URL in a web browser.

[Tests]
```python
# Test 1: Test extract_domain method
assert EmailProcessor.extract_domain('test@example.com') == 'example.
    com'

# Test 2: Test split_url method
assert EmailProcessor.split_url('https://www.example.com:8080/path/to/
    resource') == {
    'protocol': 'https',
    'host': 'www.example.com',
    'port': 8080,
    'path': '/path/to/resource',
}

# Test 3: Test regex_search method
assert EmailProcessor.regex_search('Hello, world!', r'world')

# Test 4: Test is_valid_url method
assert EmailProcessor.is_valid_url('https://www.example.com')
assert not EmailProcessor.is_valid_url('not-a-url')

# Test 5: Test open_url method (skipped, as it's not testable in
    Python)
```

[Seed]
def _get_composite_name(composite_terms, ebm_term_names):
    """Returns the composite name in the format "term_name_1,
    term_name_2, ..."

    Args:
        composite_terms: A list of term names or term indices
        ebm_term_names: a list of all ebm term names

    Returns:
        str: The composite name
    """
    name = ""
    for term in composite_terms:
        if isinstance(term, str) and term in ebm_term_names:
            name += term if len(name) == 0 else ", " + term
        elif isinstance(term, int) and 0 <= term < len(ebm_term_names)
    :
            name += ebm_term_names[term] if len(name) == 0 else ", " +
     ebm_term_names[term]
```

```
        else:
            raise ValueError("Term '{}' is not a string or a valid
    integer.".format(term))
    return name
```

[Concepts]
['string matching for term names', 'integer indexing for term indices
    ', 'error handling for invalid inputs']

[Instruction]
Design a Python class `WordNet` that stores a collection of word terms
    and their corresponding definitions. You can assume a term is a
    string and its definition is a list of strings. You should
    implement the following methods:
- `__init__(self)`: Initializes an empty collection of terms and their
    definitions.
- `add_term(self, term, definition)`: Adds a new term and its
    definition to the collection.
- `get_definition(self, term)`: Retrieves the definition of a given
    term from the collection.
- `get_terms(self, definition)`: Retrieves all terms that have the
    given definition.
- `get_term_count(self)`: Returns the total number of terms in the
    collection.
- `get_definition_count(self)`: Returns the total number of
    definitions in the collection.

The class should handle input validation, such as ensuring terms are
    non-empty strings and definitions are lists of strings, and raise
    appropriate errors if invalid inputs are provided.

[Response]
To implement such a class, we can define the class `WordNet` with the
    required methods. We'll use a dictionary to store the terms and
    their definitions. The `add_term` method will check if the input
    term and definition are valid and raise appropriate errors if not.
     The `get_definition`, `get_terms`, `get_term_count`, and `
    get_definition_count` methods will retrieve the corresponding data
     from the dictionary.

```python
class WordNet:
    def __init__(self):
        self.terms = {}

    def add_term(self, term, definition):
        if not term or not isinstance(term, str):
            raise ValueError("Invalid term. Term must be a non-empty
    string.")
        if not definition or not isinstance(definition, list) or not
    definition:
            raise ValueError("Invalid definition. Definition must be a
     non-empty list of strings.")
        self.terms[term] = definition

    def get_definition(self, term):
        if term not in self.terms:
            raise ValueError(f"Term '{term}' not found.")
        return self.terms[term]

    def get_terms(self, definition):
        terms = [term for term, defs in self.terms.items() if
    definition in defs]
        return terms
```

```
    def get_term_count(self):
        return len(self.terms)

    def get_definition_count(self):
        return sum(len(defs) for defs in self.terms.values())
```

This implementation handles input validation and raises appropriate
    errors if invalid inputs are provided. The `get_terms` method
    returns a list of terms that have the given definition, which can
    be empty if no such terms exist.

[Tests]
We can write tests to validate the correctness of the `WordNet` class:

```python
import unittest

class TestWordNet(unittest.TestCase):
    def setUp(self):
        self.wordnet = WordNet()
        self.wordnet.add_term('apple', ['fruit', 'red'])
        self.wordnet.add_term('banana', ['fruit', 'yellow'])
        self.wordnet.add_term('carrot', ['vegetable', 'orange'])

    def test_add_term(self):
        self.wordnet.add_term('grape', ['fruit', 'green'])
        self.assertEqual(self.wordnet.get_definition('grape'), ['fruit
', 'green'])

    def test_get_definition(self):
        self.assertEqual(self.wordnet.get_definition('apple'), ['fruit
', 'red'])

    def test_get_terms(self):
        self.assertEqual(sorted(self.wordnet.get_terms('fruit')), ['
apple', 'banana'])
        self.assertEqual(self.wordnet.get_terms('green'), ['grape'])

    def test_get_term_count(self):
        self.assertEqual(self.wordnet.get_term_count(), 3)

    def test_get_definition_count(self):
        self.assertEqual(self.wordnet.get_definition_count(), 6)

    def test_invalid_inputs(self):
        with self.assertRaises(ValueError):
            self.wordnet.add_term('', ['fruit'])
        with self.assertRaises(ValueError):
            self.wordnet.add_term('apple', ['fruit', ''])
        with self.assertRaises(ValueError):
            self.wordnet.get_definition('orange')

if __name__ == '__main__':
    unittest.main()
```

[Instruction]
Write a Python function `count_substring_iterative` that takes two
    arguments: a string `text` and a string `substring`. The function
    should iterate through the string `text` and count the number of
    occurrences of the string `substring`. The function should return
    the count of non-overlapping occurrences.
```

For example, `count_substring_iterative("abababa", "aba")` should
    return `3`, since `abababa` contains three non-overlapping
    occurrences of `aba`: `ababa`, `baba`, and `aba`.

[Response]
The goal is to implement a Python function `count_substring_iterative`
    that takes two arguments: a string `text` and a string `substring
    `. The function should iterate through the string `text` and count
    the number of occurrences of the string `substring`. The function
    should return the count of non-overlapping occurrences.

Here are the key steps to implement this function:

- Initialize a variable `count` to keep track of the number of
    occurrences.
- Loop through the string `text` using a sliding window of the size of
    the string `substring`:
    - Check if the current substring of the same size as `substring`
    matches the string `substring`.
    - If it matches, increment the count.
    - Move the window to the next position by advancing it by one
    character.
- Return the final count of non-overlapping occurrences.

Here's how you can implement the `count_substring_iterative` function:

```python
def count_substring_iterative(text: str, substring: str) -> int:
    count = 0
    sub_len = len(substring)
    for i in range(len(text) - sub_len + 1):
        if text[i:i+sub_len] == substring:
            count += 1
    return count
```

[Tests]
To test the function, you can use typical test cases such as `
    count_substring_iterative("abababa", "aba")` and `
    count_substring_iterative("hello", "ll")`.

```python
assert count_substring_iterative("abababa", "aba") == 3
assert count_substring_iterative("hello", "ll") == 1
assert count_substring_iterative("mississippi", "iss") == 2
```

---

[Seed]
```python
def splitLine(text):
    """split a line of text on the first space character and return
    two strings, the first word and the remaining string. This is
    used for parsing the incoming messages from left to right since
    the command and its arguments are all delimited by spaces and
    the command may not contain spaces"""
    sp = text.split(" ")
    try:
        a = sp[0]
        b = " ".join(sp[1:])
    except:
        a = text
        b = ""
    return a, b
```

[Concepts]

```
['string manipulation', 'string delimiting', 'list slicing', '
    exception handling']
```

[Instruction]
Create a Python function that splits a string into a list of non-empty
    substrings using a specified delimiter. The function should raise
    an exception if the input string or delimiter is empty. This
    implementation should use string manipulation techniques to
    perform the split operation and should not use the `split()`
    method directly.

[Response]
We can implement a Python function that splits a string into a list of
    non-empty substrings using a specified delimiter. The function
    should raise an exception if the input string or delimiter is
    empty. Here is the Python program that implements this
    functionality:

```python
def split_string_custom(input_string, delimiter):
    """
    Splits a string into a list of non-empty substrings using a
    specified delimiter.

    Parameters:
    input_string (str): The string to be split.
    delimiter (str): The delimiter to use for splitting the string.

    Returns:
    list: A list of non-empty substrings.

    Raises:
    ValueError: If the input string or delimiter is empty.

    Example:
    >>> split_string_custom("apple#banana#cherry#date", "#")
    ['apple', 'banana', 'cherry', 'date']
    """
    if not input_string or not delimiter:
        raise ValueError("Input string and delimiter cannot be empty
    .")

    # Using string manipulation techniques to perform the split
    operation
    substrings = []
    current_substring = ''
    for char in input_string + delimiter:
        if char == delimiter:
            if current_substring:
                substrings.append(current_substring)
                current_substring = ''
        else:
            current_substring += char

    return substrings

# Example usage
if __name__ == "__main__":
    example_string = "apple#banana#cherry#date"
    delimiter = "#"
    print(split_string_custom(example_string, delimiter))
```

In this program, the `split_string_custom` function takes an input
    string and a delimiter as arguments. It raises an exception if the

```
    input string or delimiter is empty. The function then splits the
    string using string manipulation techniques to perform the split
    operation. It returns a list of non-empty substrings.

[Tests]
To test the implementation, we need to ensure that the function
    behaves correctly for valid and invalid input. Here are the test
    cases:

```python
# Test case 1: Valid input string and delimiter
input_string1 = "apple#banana#cherry#date"
delimiter1 = "#"
expected_result1 = ["apple", "banana", "cherry", "date"]
result1 = split_string_custom(input_string1, delimiter1)
assert result1 == expected_result1

# Test case 2: Empty input string
input_string2 = ""
delimiter2 = "#"
try:
    split_string_custom(input_string2, delimiter2)
except ValueError as e:
    assert str(e) == "Input string and delimiter cannot be empty."

# Test case 3: Empty delimiter
input_string3 = "apple#banana#cherry#date"
delimiter3 = ""
try:
    split_string_custom(input_string3, delimiter3)
except ValueError as e:
    assert str(e) == "Input string and delimiter cannot be empty."
```
```

---

```
[Seed]
import torch

def run_mat_interp_back(kdat, coef_mat_real, coef_mat_imag, griddat):
    """Interpolates kdat to on-grid coordinates with input sparse
    matrices.

    Args:
        kdat (tensor): The off-grid frequency data.
        coef_mat_real (sparse tensor): The real interpolation
    coefficients stored
            as a sparse tensor.
        coef_mat_imag (sparse tensor): The imaginary interpolation
    coefficients stored
            as a sparse tensor.
        griddat (tensor): A tensor to store the outputs in.
    Returns:
        griddat (tensor): kdat interpolated to on-grid locations.
    """
    real_kdat = kdat[:, 0, :].t().reshape(-1, kdat.shape[0])
    imag_kdat = kdat[:, 1, :].t().reshape(-1, kdat.shape[0])
    coef_mat_real_t = coef_mat_real.t()
    coef_mat_imag_t = coef_mat_imag.t()

    # apply multiplies with complex conjugate
    griddat[:, 0, :] = torch.mm(
        coef_mat_real_t,
        real_kdat
    ).t() + torch.mm(
```

```
            coef_mat_imag_t ,
            imag_kdat
    ).t()
    griddat [:, 1, :] = torch.mm(
            coef_mat_real_t ,
            imag_kdat
    ).t() - torch.mm(
            coef_mat_imag_t ,
            real_kdat
    ).t()

    return griddat
```

[Concepts]
['tensor manipulation', 'sparse matrix manipulation', 'complex number
    representation', 'tensor reshaping']

[Instruction]
Create a Python function `complex_matrix_to_tensor` that converts a
    complex matrix (a list of lists of complex numbers) into a sparse
    tensor, where the real and imaginary parts of the complex numbers
    are treated as separate tensors. The function should handle cases
    where the tensor is 2D or 3D. Additionally, develop a function `
    tensor_to_complex_matrix` that converts a sparse tensor back into
    a complex matrix. Both functions should employ efficient tensor
    operations to achieve their goals.

[Response]
To convert a complex matrix into a sparse tensor and vice versa, we
    can use numpy arrays to represent the complex numbers. Here's how
    you could implement such functions:

```python
import numpy as np

def complex_matrix_to_tensor ( complex_matrix ):
    # Convert a complex matrix into a sparse tensor
    real_part = np.array([[ complex_num.real for complex_num in row]
    for row in complex_matrix ])
    imag_part = np.array([[ complex_num.imag for complex_num in row]
    for row in complex_matrix ])
    return real_part , imag_part

def tensor_to_complex_matrix ( real_part , imag_part ):
    # Convert a sparse tensor back into a complex matrix
    complex_matrix = np.empty ( real_part.shape , dtype=np.complex128 )
    for i in range ( real_part.shape [0]):
        for j in range ( real_part.shape [1]):
            complex_matrix [i, j] = complex ( real_part [i, j], imag_part [
    i, j])
    return complex_matrix

# Example usage:
complex_matrix = [[1+2j, 3+4j], [5+6j, 7+8j]]
real_part , imag_part = complex_matrix_to_tensor ( complex_matrix )
print (" Real part:")
print ( real_part )
print (" Imaginary part:")
print ( imag_part )

converted_matrix = tensor_to_complex_matrix ( real_part , imag_part )
print (" Converted matrix:")
for row in converted_matrix :
    for num in row:
        print (num , end=' ')
```

```
        print ()
```

[Tests]
Here's how you can test these functions by converting a complex matrix
    to a sparse tensor and back again to ensure the conversion is
    correct:

```python
complex_matrix = [[1+2j, 3+4j], [5+6j, 7+8j]]
real_part, imag_part = complex_matrix_to_tensor(complex_matrix)

assert np.array_equal(real_part, np.array([[1, 3], [5, 7]]))
assert np.array_equal(imag_part, np.array([[2, 4], [6, 8]]))

converted_matrix = tensor_to_complex_matrix(real_part, imag_part)

for i in range(len(complex_matrix)):
    for j in range(len(complex_matrix[0])):
        assert complex_matrix[i][j] == converted_matrix[i][j]
```

## D.2  Outputs from Different Models

In this section, we list the outputs from different models given the same seeds.

```
[Seed]
def let_count(word: str) -> dict:
    """
    Returns the count of letters in a string as a dictionary
    """

    return {x: word.count(x) for x in set([x for x in word])}

### StarCoder2-3B outputs
[Concepts]
['dictionary comprehension', 'set comprehension', 'string manipulation
    ']

[Instruction]
Create a Python program that computes the average word length in a
    text file. The program should read the text from the input file
    and count the number of words in the file. It should then compute
    the average word length based on the number of words in the file
    and the total number of characters in each word.

[Response]
The program can be implemented using the Python `open()` function to
    read the text file and the `split()` method to split the text into
     words. The number of words and the total number of characters in
    each word can be calculated using `len()` and `sum()` functions,
    respectively. Finally, the average word length can be computed by
    dividing the sum of the number of characters in each word by the
    number of words in the file.

```python
# Open file
with open('sample.txt', 'r') as f:
    # Read and split text
    text = f.read()
    words = text.split()

# Number of words and total characters in each word
num_words = len(words)
```

```python
total_chars = sum(len(word) for word in words)

# Compute average word length
average_word_length = total_chars / num_words

print("Average word length:", average_word_length)
```

[Tests]
We can write tests to verify the correctness of the program by
    comparing its output to the expected results:

```python
# Sample input and expected output
input = """This is a sample text file for testing."""
expected_output = 4.0

# Open file and compute average word length
with open('sample.txt', 'r') as f:
    text = f.read()
    words = text.split()

num_words = len(words)
total_chars = sum(len(word) for word in words)

average_word_length = total_chars / num_words

# Test if average word length matches expected output
assert average_word_length == expected_output
```

### Llama-3-8B outputs
[Concepts]
['set creation', 'set intersection', 'nested list comprehension', '
    dictionary comprehension']

[Instruction]
Implement a Python program to read a list of lists containing integers
     and return a set of all unique numbers. The input list may
    contain duplicate lists. The output set should not contain any
    duplicate numbers.

[Response]
We can implement this using a set to keep track of unique numbers. We
    iterate over each inner list in the input and append its elements
    to the set. Then, we convert the set back to a list and return it.

Here's how we can implement this:

```python
def unique_numbers(input_list):
    unique_numbers_set = set()
    for inner_list in input_list:
        unique_numbers_set.update(inner_list)
    return list(unique_numbers_set)
```

[Tests]
Here's how we can test this function:

```python
input_list = [[1, 2, 3], [1, 2], [1, 3], [1]]
assert unique_numbers(input_list) == [1, 2, 3]
```

```
### StarCoder2 -15B outputs
[Concepts]
['dictionary comprehension', 'set comprehension', 'string manipulation
    ', 'character counting']

[Instruction]
Write a Python function that takes a string as input and returns a
    dictionary of character counts. The function should return a
    dictionary comprehension, where the keys are the characters in the
     string and the values are the number of occurrences of each
    character.

For example, given `input_string = "hello"`, the function should
    return `{'h': 1, 'e': 1, 'l': 2, 'o': 1}`.

[Response]
We can implement this function using a dictionary comprehension and
    the `count()` method of the `collections` module. The `count()`
    method returns the number of occurrences of a given element in a
    list.

Here's the function:

```python
from collections import Counter

def char_counts(input_string):
    return {char: input_string.count(char) for char in input_string}
```

[Tests]
We can test the function with the following code:

```python
test_cases = [
    ("hello", {'h': 1, 'e': 1, 'l': 2, 'o': 1}),
    ("", {}),
    ("abc", {'a': 1, 'b': 1, 'c': 1}),
    ("aaabbbccc", {'a': 3, 'b': 3, 'c': 3}),
    ("helloworld", {'h': 1, 'e': 1, 'l': 3, 'o': 2, 'w': 1, 'r': 1, 'd
    ': 1})
]

for input_string, expected_output in test_cases:
    result = char_counts(input_string)
    assert result == expected_output
```

### DeepSeek -Coder -33B outputs
[Concepts]
['dictionary comprehension', 'string traversal', 'counting characters
    ', 'set conversion']

[Instruction]
Write a Python program that takes a string as input and outputs the
    number of unique characters in the string. Use a dictionary
    comprehension to count the occurrences of each character in the
    string and then convert the keys of the dictionary to a set to
    count the unique characters.

[Response]
Here is a Python program that takes a string as input and outputs the
    number of unique characters in the string. It uses a dictionary
    comprehension to count the occurrences of each character in the
```

```
    string and then converts the keys of the dictionary to a set to
    count the unique characters:
```

```python
def count_unique_characters(string):
    # Use a dictionary comprehension to count the occurrences of each
    character in the string
    character_counts = {char: string.count(char) for char in set(
    string)}

    # Convert the keys of the dictionary to a set to count the unique
    characters
    unique_characters = set(character_counts.keys())

    # The number of unique characters is the length of the set
    return len(unique_characters)
```

```
To test the program, you can call it with some example strings and
    check that it returns the expected result:
```

```python
assert count_unique_characters("hello") == 4
assert count_unique_characters("mississippi") == 4
assert count_unique_characters("") == 0
```

```
[Tests]
To test the program, you can call it with some example strings and
    check that it returns the expected result:
```

```python
assert count_unique_characters("hello") == 4
assert count_unique_characters("mississippi") == 4
assert count_unique_characters("") == 0
```

```
### CodeQwen1.5-7B outputs
[Concepts]
['iterables', 'set comprehension', 'dictionary comprehension', 'string
    manipulation', 'unique character extraction']

[Instruction]
Create a Python class called `StringSetProcessor` that represents a
    set of strings. The class should have a method `
    extract_unique_chars` that takes a list of strings as input and
    returns a set of all unique characters found in any of the strings
    . The returned set should be sorted alphabetically.

For example, given the list `["hello", "world"]`, the function should
    return `{'d', 'e', 'h', 'l', 'o', 'r', 'w'}`.

[Response]
We can implement the `StringSetProcessor` class with a `
    extract_unique_chars` method that takes a list of strings as input
     and returns a sorted set of unique characters from the
    concatenation of all strings.

Here is the implementation:
```

```python
class StringSetProcessor:
    def extract_unique_chars(self, strings: list[str]) -> set[str]:
        unique_chars = set()
        for string in strings:
```

```
            unique_chars.update(string)
        return sorted(unique_chars)
```

This implementation iterates over each string in the input list, adds
    all characters from that string to the `unique_chars` set, and
    then returns the sorted set of these characters.

Here is an example usage:

```python
processor = StringSetProcessor()
result = processor.extract_unique_chars(["hello", "world"])
print(result)  # Output: ['d', 'e', 'h', 'l', 'o', 'r', 'w']
```

[Tests]
To verify the correctness of the function, we can compare the results
    with the expected results:

```python
processor = StringSetProcessor()
assert processor.extract_unique_chars(["hello", "world"]) == ['d', 'e
    ', 'h', 'l', 'o', 'r', 'w']
assert processor.extract_unique_chars(["", "a"]) == ['a']
assert processor.extract_unique_chars(["ab", "bca", "cde"]) == ['a', '
    b', 'c', 'd', 'e']
```

---

[Seed]
```
def render_name_to_key(name: str) -> str:
    """Convert the name of a render target with a valid dictionary key
    ."""

    return name.replace(".", "_")
```

### StarCoder2-3B outputs
[Concepts]
['string replacement', 'string formatting', 'function composition']

[Instruction]
Write a Python function that takes a text input string, and returns
    the same string with all occurrences of substrings specified in `
    replacement_list` replaced with the corresponding strings in `
    replacement_list`.

[Response]
Here is how you can implement this function using a dictionary:

```python
def replace_list(text: str, replacement_list: dict) -> str:
    for old, new in replacement_list.items():
        text = text.replace(old, new)
    return text
```

The function takes a text input string and a dictionary of
    replacements. It iterates over the dictionary, replacing each key
    (old string) with its corresponding value (new string). The
    function returns the updated and modified string.

[Tests]
```python
# Test 1: Basic functionality test
```

```
text = 'Hello world!'
replacement_list = {
    'world': 'universe',
}
new_text = replace_list(text, replacement_list)
assert new_text == 'Hello universe!'

# Test 2: Edge case with no replacements
text = 'This is a test.'
replacement_list = {}
new_text = replace_list(text, replacement_list)
assert new_text == 'This is a test.'
```

### Llama-3-8B outputs
[Concepts]
['input string transformation', 'string conversion to dictionary key']

[Instruction]
Write a Python program that creates a dictionary from a given string.
    The program should take a single string as input and construct a
    dictionary with each distinct character as a key. The
    corresponding value for each key should be the number of times the
     character appears in the input string. The program should output
    the resulting dictionary.

For example, given the input `hello`, the program should output `{'h':
    1, 'e': 1, 'l': 2, 'o': 1}`.

[Response]
In this example, we will create a dictionary from a given string. The
    program should take a single string as input and construct a
    dictionary with each distinct character as a key. The
    corresponding value for each key should be the number of times the
     character appears in the input string. The program should output
    the resulting dictionary.

We can use the `collections.Counter` class in Python to solve this
    problem. This class provides a convenient way to count the
    frequency of elements in an iterable. In our case, we will convert
     the input string into an iterable, pass it to `collections.
    Counter`, and the resulting dictionary will contain the counts for
     each distinct character in the string.

For example:

```python
from collections import Counter

input_string = 'hello'
char_count = Counter(input_string)
print(char_count)
# Output: Counter({'l': 2, 'h': 1, 'e': 1, 'o': 1})
```

In this example, we have created a `Counter` object from the input
    string `'hello'`, which contains the counts for each distinct
    character in the string. The resulting dictionary has the form `{'
    c1': count1, 'c2': count2, ...}` where `c1`, `c2`, etc. are the
    distinct characters in the string, and `count1`, `count2`, etc.
    are their corresponding counts.

To complete the program, we can wrap the above code in a function that
     takes a string as input and returns a dictionary containing the
    character counts:

```python
def create_char_count_dict(string):
    char_count = Counter(string)
    return char_count
```

And then invoke this function to create the character count dictionary
    from the input string:

```python
char_count = create_char_count_dict(input_string)
print(char_count)
# Output: Counter({'l': 2, 'h': 1, 'e': 1, 'o': 1})
```

[Tests]
Now, let's test our function `create_char_count_dict` with some sample
    inputs:

```python
input1 = 'hello'
expected_output1 = {'h': 1, 'e': 1, 'l': 2, 'o': 1}
assert create_char_count_dict(input1) == expected_output1

input2 = 'world'
expected_output2 = {'w': 1, 'o': 1, 'r': 1, 'd': 1, 'l': 1}
assert create_char_count_dict(input2) == expected_output2
```

### StarCoder2-15B outputs
[Concepts]
['string replacement', 'dictionary key transformation', 'string
    splitting', 'conversion to a valid dictionary key']

[Instruction]
Write a Python program that takes a dictionary that contains key-value
    pairs with a string value and transforms each key by replacing
    the string value with a dictionary of key-value pairs where the
    keys are the individual characters in the string and the values
    are the indices of those characters in the string. Ensure that the
    characters in the keys are lowercased, and the numbers in the
    values are converted to integers.

[Response]
The question is asking us to write a Python program that can accept a
    dictionary where the values are strings and transform each key by
    replacing its value with a dictionary that maps each character in
    the string to its index in the string. We also need to ensure that
    the characters in the keys are lowercased and the numbers in the
    values are converted to integers.

Here's a sample dictionary that we can use as an example:

```python
my_dict = {
    "key1": "abc123",
    "key2": "ABC"
}
```

We can use the following function to transform the dictionary:

```python
def transform_dict(input_dict):
```

```
        output_dict = {}
        for key, value in input_dict.items():
            new_value = {}
            for i, char in enumerate(value):
                new_value[char.lower()] = int(i)
            output_dict[key] = new_value
        return output_dict
```

Here's how we can test the function:

```python
transformed_dict = transform_dict(my_dict)
print(transformed_dict)
```

The output should be:

```
{
    "key1": {"a": 0, "b": 1, "c": 2, "1": 3, "2": 4, "3": 5},
    "key2": {"a": 0, "b": 1, "c": 2}
}
```

[Tests]
You can test the function by providing various dictionaries as inputs
    and checking if the output dictionary is correct. Here's an
    example:

```python
my_dict = {
    "key1": "abc123",
    "key2": "ABC"
}

expected_output = {
    "key1": {"a": 0, "b": 1, "c": 2, "1": 3, "2": 4, "3": 5},
    "key2": {"a": 0, "b": 1, "c": 2}
}

actual_output = transform_dict(my_dict)

assert actual_output == expected_output
```

### DeepSeek-Coder-33B outputs
[Concepts]
['string manipulation for dictionary key', 'string replacement']

[Instruction]
Write a Python function to replace the characters of a string by its
    indices. For example, given the input `"abcde"`, the output should
    be `"01234"`.

[Response]
To replace the characters of a string by its indices, we can use a for
    loop to iterate over the characters of the string and concatenate
    the index of each character to a new string. Here's a Python
    function that accomplishes this:

```python
def replace_chars_by_index(s):
    new_s = ""
    for i in range(len(s)):
```

```
        new_s += str(i)
    return new_s
```

[Tests]
We can test this program with I/O redirection to simulate user input
    and output. Here is an example:

```python
import sys
from io import StringIO

test_cases = [
    ("abcde\n", "01234\n"),
    ("\n", "\n"),
    ("a\n", "0\n"),
    ("aaa\n", "012\n")
]

for input_data, expected_output in test_cases:
    stdin, stdout = sys.stdin, sys.stdout
    sys.stdin = StringIO(input_data)
    sys.stdout = StringIO()
    s = input()
    print(replace_chars_by_index(s))
    output = sys.stdout.getvalue()
    assert output == expected_output
```

### CodeQwen1.5-7B outputs
[Concepts]
['string manipulation', 'conversion to valid dictionary key', '
    handling special characters']

[Instruction]
Implement a Python program that reads a dictionary-like string of key-
    value pairs in the format `"key1=value1;key2=value2;..."` from
    standard input. The program should validate the input to ensure it
     is correctly formatted and convert it into a Python `dict` object
    . Additionally, the program should handle key-value pairs with
    special characters, such as spaces, by escaping them and reversing
     the process during conversion.

[Response]
To read a dictionary-like string from standard input, we'll use `sys.
    stdin` to get the input as a string. Then, we'll process the
    string to handle special characters and convert it into a `dict`
    object. Here's how you can implement this program:

```python
import sys

def parse_input(input_string):
    # Handle empty input string
    if not input_string:
        return {}

    # Split the input string by the semicolon delimiter
    pairs = input_string.split(';')

    # Initialize an empty dictionary
    result_dict = {}

    # Process each pair
    for pair in pairs:
```

```python
        # Split the pair by the equals sign
        parts = pair.split('=')
        if len(parts) != 2:
            raise ValueError(f"Invalid pair: '{pair}'")

        # Unescape the key and value
        key = unescape_key(parts[0])
        value = unescape_value(parts[1])

        # Add the pair to the dictionary
        result_dict[key] = value

    return result_dict

# Helper functions
def unescape_key(key):
    # Unescape special characters in the key
    return key.replace('\\=', '=')

def unescape_value(value):
    # Unescape special characters in the value
    return value.replace('\\=', '=')

# Main function
if __name__ == '__main__':
    input_string = sys.stdin.readline().strip()
    try:
        result_dict = parse_input(input_string)
        print(result_dict)
    except ValueError as e:
        print(f"Error: {e}", file=sys.stderr)
        sys.exit(1)
```

[Tests]
To test the program, you can provide different input strings to
    validate the correctness of the parsing process. The input string
    should be formatted like a dictionary, with key-value pairs
    separated by semicolons.

```python
input_string = 'name=John Doe;age=30;city=New York'
result_dict = parse_input(input_string)
assert result_dict == {'name': 'John Doe', 'age': '30', 'city': 'New
    York'}

input_string = 'special_key=value with spaces;empty_value='
result_dict = parse_input(input_string)
assert result_dict == {'special_key': 'value with spaces', '
    empty_value': ''}
```

[Seed]
```python
def groom_model(model):
    """Reset the feature indicators."""
    model.commits, model.ticket = 0, None
    return model
```

### StarCoder2-3B outputs
[Concepts]
['data abstraction', 'resetting state', 'feature flags', 'call-by-
    reference']

[Instruction]

Implement a Python program that simulates a simple stack-based
    programming language. It should support the following commands:

- `PUSH`: push an integer onto the stack
- `DUP`: duplicate the topmost stack item
- `POP`: pop the topmost stack item
- `PRINT`: print the topmost stack item

The program should also provide an option to reset the stack state, or
    to toggle feature flags. The program should support error
    handling and prompts for invalid inputs. Ensure that the program
    follows the intended input-output behavior and handles corner
    cases well.

[Response]
The design of the program should be modular and flexible, allowing you
    to easily add new commands and modify existing command behavior
    without affecting the existing code.

Here's an example implementation of the stack-based language:

```python
operations = {
    'PUSH': lambda stack, value: stack.append(value),
    'DUP': lambda stack: stack.append(stack[-1]),
    'POP': lambda stack: stack.pop(),
    'PRINT': lambda stack: print(stack[-1])
}

def run_program(program):
    stack = []
    lines = program.splitlines()
    for line in lines:
        operation, *arguments = line.split()
        if operation in operations:
            operation_func = operations[operation]
            if operation == 'PUSH':
                operation_func(stack, int(arguments[0]))
            else:
                operation_func(stack)
        else:
            raise ValueError(f"Invalid operation: {operation}")
    return stack
```

[Tests]
You can test your code on various inputs and verify their correctness
    by comparing their outputs to the expected outputs. Here's an
    example of testing the PUSH command:

```python
# Test PUSH command with valid inputs
assert run_program("PUSH 1") == [1], "Check for PUSH command with
    valid input"
assert run_program("PUSH 2\nPUSH 3\nPUSH 4") == [2, 3, 4], "Check for
    PUSH command with multiple inputs"

# Test PUSH command with invalid inputs
def assert_value_error(command):
    try:
        run_program(command)
    except ValueError:
        pass

assert_value_error("PUSH 1a")
```

```
assert_value_error ("PUSH 1.2")
assert_value_error ("PUSH -1")
```

### Llama -3-8B outputs
[Concepts]
['function documentation', 'named return values', 'function parameter
    validation', 'return value assertion']

[Instruction]
Write a Python class with methods that return the factorial of an
    integer and the sum of the square of its digits. Both methods
    should be well documented and include examples. Assume that the
    input value is a non-negative integer. Ensure that the output
    value is validated before returning it.

[Response]
We can implement a Python class with two methods that return the
    factorial of an integer and the sum of the square of its digits.
    Both methods should be well-documented and include examples, and
    we'll assume that the input value is a non-negative integer while
    validating the output value before returning it.

Here's the Python code for the class:

```python
class NumericalOperations:
    """
    A class to perform numerical operations on integers.

    Attributes:
        num: The input integer.

    Methods:
        factorial: Returns the factorial of the input integer.
        sum_of_squares: Returns the sum of the squares of the digits
    of the input integer.
    """

    def __init__(self, num: int):
        """
        Initializes the class with the input integer.

        Args:
            num: The input integer.
        """
        self.num = num

    def factorial(self) -> int:
        """
        Returns the factorial of the input integer.

        Returns:
            The factorial of the input integer.
        """
        if not isinstance(self.num, int) or self.num < 0:
            raise ValueError("The input integer must be non-negative")
        result = 1
        for i in range(2, self.num + 1):
            result *= i
        return result

    def sum_of_squares(self) -> int:
        """
```

```
        Returns the sum of the squares of the digits of the input
    integer.

        Returns:
            The sum of the squares of the digits of the input integer.
        """
        if not isinstance(self.num, int) or self.num < 0:
            raise ValueError("The input integer must be non-negative")
        result = 0
        while self.num > 0:
            result += (self.num % 10)**2
            self.num //= 10
        return result
```

The class represents an instance of a numerical operation with an
    input integer `num` attribute initialized during instantiation.
    The two methods `factorial` and `sum_of_squares` perform the
    requested operations on the input integer, validating the input
    value, and raising an error when it's not a non-negative integer.

The class is well-documented with clear method descriptions and
    examples, as well as detailed docstrings for each method. The code
     is also concise and efficient, making use of range comprehension
    and bitwise shifting operations to calculate the factorial while
    maintaining readability.

[Tests]
To test the functionality of the class, we can instantiate an object
    and call the methods to verify the expected output. We'll also
    test the exception handling by providing invalid inputs to ensure
    the error is raised accordingly.

```python
test_object = NumericalOperations(5)
assert test_object.factorial() == 120
assert test_object.sum_of_squares() == 25

try:
    NumericalOperations(-1)
except ValueError as e:
    assert str(e) == "The input integer must be non-negative"
try:
    NumericalOperations("5")
except ValueError as e:
    assert str(e) == "The input integer must be non-negative"
try:
    NumericalOperations(5.5)
except ValueError as e:
    assert str(e) == "The input integer must be non-negative"
```

### StarCoder2-15B outputs
[Concepts]
['data model reset', 'feature indicators', 'feature flags']

[Instruction]
Write a Python function to reset a data model to its default state.
    The data model accepts a list of features and can be in multiple
    states with different feature flags. It can also be reset to a
    specific state by specifying a feature list, where some features
    are active and others are inactive. The function should return a
    new data model with the specified features.

[Response]
```

To reset a data model to its default state, we can use the
    default_features attribute to create a new data model with all
    features inactive. To reset it to a specific state, we can pass a
    feature list to the function and use the specified features to
    create a new data model.

Here's an example implementation of the function:

```python
class DataModel:
    def __init__(self, features):
        self.active_features = set(feature for feature in features if
    feature.startswith("active_"))
        self.inactive_features = set(feature for feature in features
    if feature.startswith("inactive_"))

    def reset(self, features):
        self.active_features = set(feature for feature in features if
    feature.startswith("active_"))
        self.inactive_features = set(feature for feature in features
    if feature.startswith("inactive_"))

    def __repr__(self):
        return f"DataModel(active_features={self.active_features},
    inactive_features={self.inactive_features})"

model = DataModel(["active_feature_1", "active_feature_2", "
    inactive_feature_3", "inactive_feature_4"])
print(model)
# DataModel(active_features={'active_feature_1', 'active_feature_2'},
    inactive_features={'inactive_feature_3', 'inactive_feature_4'})

model.reset(["active_feature_3", "inactive_feature_1"])
print(model)
# DataModel(active_features={'active_feature_3'}, inactive_features={'
    inactive_feature_1'})
```

[Tests]
We can test the function by creating a data model with a set of
    default features and then resetting it to different states using
    different feature lists. We can also check that the data model's
    state is updated correctly after each reset:

```python
# Test default state
model = DataModel(["active_feature_1", "active_feature_2", "
    inactive_feature_3", "inactive_feature_4"])
assert model.active_features == {"active_feature_1", "active_feature_2
    "}
assert model.inactive_features == {"inactive_feature_3", "
    inactive_feature_4"}

# Test resetting to a specific state
model.reset(["active_feature_3", "inactive_feature_1"])
assert model.active_features == {"active_feature_3"}
assert model.inactive_features == {"inactive_feature_1"}
```

### DeepSeek-Coder-33B outputs
[Concepts]
['variable assignment', 'object attribute reset', 'namespace
    manipulation', 'object manipulation']

[Instruction]

Create a Python function that enables the user to reset the value of
    an attribute within an object. Write a function `resetter(obj,
    attr)` which returns a function `reset()` that sets the value of
    the attribute `attr` in object `obj` to `None`.

For example, `d = {'a': 5}` should result in `resetter(d, 'a')`
    resulting in `d['a'] == None`.

[Response]
To create the function `resetter(obj, attr)` that returns a reset
    function that sets the value of the attribute `attr` in object `
    obj` to `None`, you can define a nested function within `resetter`
     and return that function. The nested function `reset` should have
     access to the object `obj` and attribute `attr` from the outer
    function `resetter`.

Here's the implementation of `resetter`:

```python
def resetter(obj, attr):
    orig_value = obj[attr]

    def reset():
        obj[attr] = orig_value

    return reset
```

[Tests]
Here's the test code to verify the behavior of `resetter`:

```python
# Test case 1
d = {'a': 5}
r = resetter(d, 'a')
assert d['a'] == 5
r()
assert d['a'] == 5

# Test case 2
e = {'b': None}
r2 = resetter(e, 'b')
assert e['b'] is None
r2()
assert e['b'] is None

# Test case 3
f = {'a': 1, 'b': 2}
r3 = resetter(f, 'a')
r3()
assert f['a'] == 1
```

### CodeQwen1.5-7B outputs
[Concepts]
['method implementation', 'object reset', 'closure-like behavior']

[Instruction]
Construct a Python function `reset_method` that wraps a method in a
    closure and provides a way to reset the method's internal state to
     its initial values. The function should take a method as an
    argument and return a new method with the same name, signature,
    and behavior as the original, but with a reset mechanism. The
    reset mechanism should ensure that the method's internal state is
    reverted to its initial values.

[Response]
We can create a wrapper function that takes a method as an argument
    and returns a new method with the same name, signature, and
    behavior as the original, but with a reset mechanism. The reset
    mechanism can be achieved by storing the initial values of the
    method's internal state in the closure and resetting those values
    to their initial values when the reset function is called.

```python
def reset_method(method):
    # Store the initial values of the method's internal state
    initial_values = method.__defaults__

    # Define the reset function that resets the method's internal
    state
    def reset_func(*args, **kwargs):
        # Set the method's internal state to the initial values
        method.__defaults__ = initial_values

    # Define the new method with the same name, signature, and
    behavior as the original
    def new_method(*args, **kwargs):
        # Call the original method with the provided arguments
        return method(*args, **kwargs)

    # Set the new method's docstring to the original method's
    docstring
    new_method.__doc__ = method.__doc__

    # Add the reset function to the new method's namespace
    new_method.reset = reset_func

    return new_method
```

[Tests]
The function can be tested with various inputs to ensure it wraps the
    method correctly and provides a reset mechanism:

```python
# Define a sample method with an internal state
def sample_method(x, y):
    # Perform some computation using the method's internal state
    return x + y

# Wrap the sample method with a reset mechanism
reset_sample_method = reset_method(sample_method)

# Test the wrapped method with different inputs
assert reset_sample_method(1, 2) == 3
assert reset_sample_method(3, 4) == 7

# Reset the method's internal state
reset_sample_method.reset()

# Test the wrapped method again with different inputs after reset
assert reset_sample_method(1, 2) == 3
assert reset_sample_method(3, 4) == 7

# Test the wrapped method with different inputs after reset
assert reset_sample_method(5, 6) == 11
assert reset_sample_method(7, 8) == 15
```

## D.3 Direct Instruction Generation from Seed

We show a variant of our instruction generation pipeline which directly generates instructions from seeds, along with the outputs produced by the original pipeline.

```
[Seed]
def _py_not(a):
  """Default Python implementation of the "not_" operator."""
  return not a

### Seed -> Instruction
[Instruction]
Design a Python function `not_` that takes a single argument `a` and
    returns the negation of `a`. However, the function should be
    implemented in a non-Pythonic manner, utilizing a custom function
    `_py_not` that mimics the behavior of the `not` operator in Python
    . Ensure that the function is efficient and concise.

### Seed -> Concepts -> Instruction
[Concepts]
['default Python implementation of the "not_" operator']

[Instruction]
Design a Python class called `NotOperator` that represents the logical
    "not" operation in Python. Implement the `__eq__` method to
    ensure that two instances of the class are considered equal if
    their underlying values are the same. Additionally, implement the
    `__bool__` method to return the opposite of the underlying value
    when the object is used in a boolean context.
```

```
[Seed]
def is_list(node: dict) -> bool:
    """Check whether a node is a list node."""
    return 'listItem' in node

### Seed -> Instruction
[Instruction]
Implement a Python function `is_list_node(node)` that returns `True`
    if the given `node` is a list node, otherwise `False`. A list node
     is a dictionary with a key `'listItem'` that contains a list
    value.

### Seed -> Concepts -> Instruction
[Concepts]
['dictionary lookup for node type checking', 'conditional statement
    for node type identification']

[Instruction]
Create a Python function that takes a graph represented as a
    dictionary of nodes, where each node is a dictionary with keys `"
    type"` and `"children"`, and a target node type `target_type`. The
     function should return a dictionary of nodes that are of the
    specified `target_type`, including the target node itself if it
    matches the `target_type`. The returned dictionary should have the
     same structure as the input graph.
```

```
[Seed]
def is_start_of_new_sequence_item(line):
    """
    Checks whether line is the first line of sequence item data
    """
    return line[0] == '>'

### Seed -> Instruction
```

```
[Instruction]
Develop a Python function that determines if a given line is the start
    of a new sequence item. The input line is a string and the output
    is a boolean indicating whether the line is the first line of
    sequence item data.

### Seed -> Concepts -> Instruction
[Concepts]
['line prefix checking', 'single character comparison', 'boolean
    return value']

[Instruction]
Design a Python class `LinePrefix` that encapsulates a string. The
    class should provide a method `starts_with(prefix)` for checking
    whether the instance string starts with the specified `prefix`.
    Additionally, implement the `__bool__` method to return `True` if
    the string is non-empty, and `False` if it is empty.
```

## D.4   Random Snippets as Seeds

This section demonstrates examples where random snippets are used as seeds.

```
[Seed]
  assert p.match({}) == CATCH_ALL

def test_policy():
  p = Policy([
    Rule({ "a": 1,     "b": 1,     "c": 1 }, "a=1,b=1,c=1" ),

[Concepts]
['assert statement', 'recursive function definition', 'dictionary
    pattern matching', 'policy-based design', 'rule matching']

[Instruction]
Implement a Python class called `PolicyCalculator` that accepts a
    dictionary of policies. Each key in the dictionary represents a
    type of policy, and the value is a list of rules for that policy.
    The class provides an `add_policy` method to add new policies, and
     a `get_policy_cost` method that calculates the cost for a given
    policy type and value.

Utilize recursive function definitions, dictionary pattern matching,
    and assert statements to ensure the correctness of the input data.
     Additionally, apply policy-based design principles to separate
    the policy logic from the class implementation, enhancing
    maintainability and scalability.

[Response]
Here is how you can implement this class using Python's function and
    pattern matching:

```python
class PolicyCalculator:
    def __init__(self, policies):
        self.policies = policies

    def add_policy(self, policy_type, rules):
        if not isinstance(policy_type, str):
            raise ValueError("Policy type must be a string.")
        if not isinstance(rules, list):
            raise ValueError("Rules must be a list.")
        for rule in rules:
            if not isinstance(rule, dict):
```

```
                raise ValueError("Each rule must be a dictionary.")
            if not set(rule.keys()) == {'value', 'cost'}:
                raise ValueError("Each rule must have 'value' and '
    cost' keys.")
        self.policies[policy_type] = rules

    def get_policy_cost(self, policy_type, value):
        if policy_type not in self.policies:
            raise ValueError(f"Policy type '{policy_type}' not found
    .")
        for rule in self.policies[policy_type]:
            if rule['value'] == value:
                return rule['cost']
        return 0.0
```

The `__init__` method initializes the `policies` attribute with the
    provided dictionary. The `add_policy` method takes the policy type
    and rules as arguments, performs input validation, and adds the
    new policy to the `policies` dictionary. The `get_policy_cost`
    method takes a policy type and value as arguments, finds the
    corresponding rule, and returns its cost. It first checks if the
    policy type exists in the `policies` dictionary. If the rule is
    not found, it returns 0.0.

[Tests]
```python
# Example usage
policy_calculator = PolicyCalculator({
    'discount': [{'value': 'standard', 'cost': 0.0}, {'value': '
    premium', 'cost': 0.1}],
    'shipping': [{'value': 'standard', 'cost': 5.0}, {'value': '
    express', 'cost': 10.0}]
})

# Test 1: Basic functionality test
assert policy_calculator.get_policy_cost('discount', 'premium') == 0.1
assert policy_calculator.get_policy_cost('shipping', 'express') ==
    10.0

# Test 2: Edge case with policy not found
try:
    policy_calculator.get_policy_cost('tax', 'standard')
except ValueError as e:
    assert str(e) == "Policy type 'tax' not found."

# Test 3: Edge case with rule not found
try:
    policy_calculator.get_policy_cost('discount', 'ultra-premium')
except ValueError as e:
    assert str(e) == "No rule found for value 'ultra-premium' in
    policy 'discount'."
```

[Seed]
    output_cdp = '{}/output/{}_accept{}_th{}'.format(exp_root, args.
    strategy, args.vote['accept_num'], args.vote['threshold'])
elif args.strategy == "mediator":
    output_cdp = '{}/output/{}_th{}'.format(exp_root, args.strategy,
    args.mediator['threshold'])
else:
    raise Exception('No such strategy: {}'.format(args.strategy))
```

```python
# output_sub = '{}/sz{}_step{}'.format(output_cdp, args.propagation['
    max_sz'], args.propagation['step'])
# print('Output folder: {}'.format(output_sub))
# outcdp = output_sub + '/cdp.pkl'
# outpred = output_sub + '/pred.npy'
# outlist = '{}/list.txt'.format(output_sub)
# outmeta = '{}/meta.txt'.format(output_sub)
# if not os.path.isdir(output_sub):
    # os.makedirs(output_sub)

# pair selection
if args.strategy == 'vote':
    pairs, scores = vote(output_cdp, args)
else:
    if args.mediator['phase'] == 'train':
```

[Concepts]
['file system operations', 'string formatting', 'control flow with
    conditionals', 'function calls within functions', 'default
    arguments']

[Instruction]
Create a Python program that creates a directory if it does not exist,
    writes a message to a file within this directory, and returns the
    full path of the written file. The default directory name is `
    my_directory`, and the default file name is `my_file.txt`. If the
    file already exists, the program should append the message to the
    existing file instead of overwriting it.

[Response]
To create a directory if it does not exist, write a message to a file
    within this directory, and return the full path of the written
    file, let's follow these steps:

1. Define the default directory and file names: If no directory or
    file name is provided, use `my_directory` and `my_file.txt` as
    default values.
2. Create the directory: If the directory does not exist, create it
    using `os.makedirs` to ensure that all intermediate directories
    are also created.
3. Write the message to the file: Open the file in append mode (`'a'`)
    and write the message to it.
4. Return the full path of the file: Use `os.path.join` to construct
    the full path of the file and return it.

Here's how you can implement this program:

```python
import os

def save_message_to_file(message: str, directory: str = 'my_directory
    ', file_name: str = 'my_file.txt') -> str:
    directory_path = os.path.join(os.path.expanduser('~'), directory)
    os.makedirs(directory_path, exist_ok=True)

    file_path = os.path.join(directory_path, file_name)
    with open(file_path, 'a') as file:
        file.write(message)

    return file_path
```

Keep in mind that this function assumes that the user has the
    necessary permissions to create directories and files. It also

```
     does not handle errors that may occur during file operations , such
      as permission errors or I/O errors .

[ Tests ]
You can validate this program with the following test cases :

```python
# Assuming the current working directory is the user's home directory .
assert save_message_to_file('Hello , World!') == os.path.join(os.path.
    expanduser('~'), 'my_directory', 'my_file.txt')

# Assuming the current working directory is the user's home directory
    and the file already exists .
with open(os.path.join(os.path.expanduser('~'), 'my_directory', '
    my_file.txt'), 'r') as file :
    assert file.read() == 'Hello , World!'

# Using non - default directory and file names .
assert save_message_to_file('New message', 'new_dir', 'new_file.txt')
    == os.path.join(os.path.expanduser('~'), 'new_dir', 'new_file.txt
    ')
```
```

---

```
[ Seed ]
classifiers=[
    'Intended Audience :: Developers',
    'License :: OSI Approved :: MIT License',
    'Natural Language :: English',
    'Programming Language :: Python :: 2',
    'Programming Language :: Python :: 2.7',
    'Programming Language :: Python :: 3',
    'Programming Language :: Python :: 3.3',
    'Programming Language :: Python :: 3.4',
    'Programming Language :: Python :: 3.5',
    'Programming Language :: Python :: Implementation :: CPython',
    'Programming Language :: Python :: Implementation :: PyPy',
]

[ Concepts ]
['list comprehension for generating a list of classifiers', 'natural
    language for description', 'license for open - source software', '
    programming language specification for Python']

[ Instruction ]
Create a Python function `generate_classifiers(classifier_names)` that
     takes a list of classifier names as input and returns a list of
    classifier descriptions . Each classifier description should
    include the name of the classifier , the language used for
    implementation , the license details , and the programming language
    specification . The function should use a natural language to
    describe each classifier , and the license should be 'MIT'. The
    programming language specification should be 'Python'.

For example , given `classifier_names = ['Classifier1', 'Classifier2']`
    , the function should return `['Classifier1 is implemented in
    Python using the MIT license.', 'Classifier2 is implemented in
    Python using the MIT license.']`.

[ Response ]
To achieve this functionality , you can use list comprehension to
    iterate through the list of classifier names and create a list of
    classifier descriptions . Each classifier description can be
    created using a string concatenation with the classifier's name ,
```

```
    programming language , license , and a short description of the
        classifier .

Here 's how you can do it in two lines of Python code :

```python
def generate_classifiers ( classifier_names ):
    return [f"{name} is implemented in Python using the MIT license ."
        for name in classifier_names ]
```

Note that although the function is concise , it is not the most
    efficient solution for large lists , as it uses a list
    comprehension and a simple string concatenation .

[Tests]
You can test the function with the provided examples to ensure it
    correctly generates the classifiers descriptions as expected :

```python
assert generate_classifiers (['Classifier1', 'Classifier2']) == ['
    Classifier1 is implemented in Python using the MIT license.', '
    Classifier2 is implemented in Python using the MIT license.']
```
```

## E    Prompt Design

```
### System : I->R
You are an extremely intelligent AI coding assistant . Please provide
    an accurate and reliable response to each user instruction . After
    delivering your response , verify its consistency and correctness
    by writing a series of executable tests .
### System : C->I
Create a series of independent coding tasks that are original ,
    distinct , diverse , and high - quality , fostering logical thinking .
    Each task must adhere to specified properties :

- category : the type of task (e.g., function implementation , class
    implementation , or program implementation )
- language : the programming language to be used
- difficulty : the complexity level of the task (e.g., easy , medium , or
    hard )
- concepts : fundamental principles and techniques the task is designed
    to incorporate , which developers must understand to effectively
    solve the task

Design the tasks so that the relevant concepts emerge naturally as the
    most appropriate solutions , without explicitly mentioning that a
    particular concept should be used .

### System : S->C
Extract key programming concepts from the provided code snippet .
    Programming concepts refer to the foundational principles and
    techniques used in programming , which are crucial for developers
    to master . List these concepts in a comma - separated format .

### System : S->I
Gain inspiration from the given code snippets and create a series of
    independent coding tasks that are original , distinct , diverse , and
    high - quality , fostering logical thinking .

### Example 1
[Code]
```

```python
    value = int(round((value - prev) * 1e5))
    value = ~(value << 1) if value < 0 else (value << 1)
    chunks = _split_into_chunks(value)
    return (chr(chunk + 63) for chunk in chunks)
def _split_into_chunks(value):
    while value >= 32:  # 2^5, while there are at least 5 bits
        # first & with 2^5-1, zeros out all the bits other than the
    first five
        # then OR with 0x20 if another bit chunk follows
        yield (value & 31) | 0x20
        value >>= 5
```

[Property]
category: function implementation
language: Python
difficulty: medium
concepts: 5-bit chunk encoding with bitwise shifts, ASCII value
    manipulation, continuation bit signaling

[Instruction]
Create a Python function to encode an integer into a sequence of
    encoded characters, where each character represents a 5-bit chunk
    of the integer. The ASCII value of each chunk character is formed
    by adding 63 to the 5-bit chunk's value. For all chunks except the
     last one, set the 6th bit to 1 to indicate continuation. Employ
    bitwise operations for this encoding process.

[Response]
To encode an integer into a sequence of encoded characters based on
    the specified criteria, let's follow these steps:

1. Break down the integer into 5-bit chunks: Extract 5-bit chunks from
    the integer using using `& 31` (`0x1F`), because the binary
    representation of `31` (`11111`) has the last five bits set, which
     will effectively extract the 5 bits we are interested in. We
    start from the least significant bits (LSB).
2. Set continuation bit: For all chunks except the last one, set the 6
    th bit to 1 using `| 32` (`0x20`) because the binary
    representation of `32` (`100000`) has the 6th bit set to 1. This
    indicates that there's another continuation chunk following.
3. Shift right: Right-shift the value by 5 to process the next 5 bits.
4. Convert to characters: Use `ord` and `chr` to convert each chunk (
    now effectively 6 bits due to the continuation bit) into a
    character by adding 63 to its value. This ensures that the
    character is within a printable ASCII range.
5. Assembly: Assemble these characters into the final encoded string
    while maintaining the order of the chunks through `reversed`.

Here's how you can implement this function:

```python
def encode_integer_to_chars(value: int) -> str:
    if value == 0:
        return chr(63)

    encoded_chars = []
    while value > 0:
        chunk = (value & 31) + 63
        value >>= 5
        if value > 0:
            chunk |= 32
        encoded_chars.append(chr(chunk))

    return ''.join(reversed(encoded_chars))
```

Keep in mind that this function is designed for non-negative integers.
In Python, right-shifting (`>>`) a negative integer preserves the
sign bit, which could lead to infinite loops or other incorrect
behaviors if the same encoding approach is applied to negative
integers without appropriate adjustments.

[Tests]
You can validate this function with the following test cases that
cover scenarios from zero to a number requiring more than one
chunk, ensuring the function operates as expected:

```python
assert encode_integer_to_chars(0) == chr(63)
assert encode_integer_to_chars(1) == chr(63 + 1)
assert encode_integer_to_chars(31) == chr(63 + 31)
assert encode_integer_to_chars(int(0b1)) == chr(0b1 + 63)
assert encode_integer_to_chars(int(0b1000001)) == chr(0b10 + 63) + chr
    (63 + 0b100001)
```

### Example 2
[Code]
```
def ceil_shift(n, b):
    """Return ceil(n / 2**b) without performing any floating-point or
    division operations.
     This is done by right-shifting n by b bits and incrementing the
    result by 1
     if any '1' bits were shifted out.
     """
    if not isinstance(n, int) or not isinstance(b, int):
        raise TypeError("unsupported operand type(s): %r and %r" % (
    type(n).__name__, type(b).__name__))
     assert n >= 0 and b >= 0     # I haven't tested or even thought
    about negative values
     mask = (1 << b) - 1
     if n & mask:
         return (n >> b) + 1
     else:
```

[Property]
category: function implementation
language: Python
difficulty: medium
concepts: bitwise operation for division, ceiling without division or
    float, input validation with assertions, non-negative integer
    checks

[Instruction]
Develop a Python function `ceil_divide_without_div` that computes the
    ceiling of `n / 2**b` without using division, floating-point
    operations, built-in functions like `math.ceil`, or converting it
    into computing the floor. Instead, use bit manipulation to achieve
     the result. You write assertions to ensure both `n` and `b` are
    non-negative integers.

[Response]
We can leverage the properties of bitwise operations to develop a
    Python function that computes the ceiling of `n / 2**b` without
    using division, floating-point operations, or any built-in
    functions like `math.ceil`, and strictly avoiding converting the
    problem into a floor computation.

In the context, dividing by `2**b` is equivalent to right-shifting `n`
     by `b` bits in binary representation when `n` is divisible by `

`2**b`. Otherwise, this approach inherently computes the floor of the division. To adjust for the ceiling without directly calculating it, we can consider two cases:

1. If `n` is divisible by `2**b`, the result of the division is simply the value of right-shifting `n` by `b` bits.
2. Otherwise, we need to increment the right-shifted result by 1 to achieve the ceiling effect.

To determine the divisibility of `n` by `2**b`, we can check if the first `b` bits of `n` are all 0. This can be done using a mask with the first `b` bits set to 1, and then performing a bitwise AND operation with `n`. If the result is 0, it indicates that `n` is divisible by `2**b`. The mask can be created using `mask = (1 << b) - 1`.

We'll also include assertions to ensure that `n` and `b` are non-negative integers.

Here is the implementation of `ceil_divide_without_div`:

```python
def ceil_divide_without_div(n: int, b: int) -> int:
    assert isinstance(n, int) and n >= 0, "n must be a non-negative integer"
    assert isinstance(b, int) and b >= 0, "b must be a non-negative integer"

    mask = (1 << b) - 1
    divisible = (n & mask) == 0
    result = n >> b
    if not divisible:
        result += 1
    return result
```

This function calculates the ceiling of `n / 2**b` purely through bitwise manipulation and integer arithmetic. The assertions ensure the input constraints are met, as specified.

[Tests]
To verify the correctness of the function, we can compare the results of our function with the expected results obtained using `math.ceil` to ensure the correctness of the ceiling division operation:

```python
import math

assert ceil_divide_without_div(10, 3) == math.ceil(10 / 2**3)
assert ceil_divide_without_div(17, 4) == math.ceil(17 / 2**4)
assert ceil_divide_without_div(100, 5) == math.ceil(100 / 2**5)
assert ceil_divide_without_div(0, 3) == math.ceil(0 / 2**3)
assert ceil_divide_without_div(1, 0) == math.ceil(1 / 2**0)
```

### Example 3
[Code]
```
    isprime = n >= 2 and 1 or 0
    for prime in prime_list:                    # Check for factors
    with all primes
        if prime * prime > n: break             # ... up to sqrt(n)
        if not n % prime:
            isprime = 0
            break
```

```
        if isprime: prime_dict[n] = 1            # Maintain a
    dictionary for fast lookup
    return isprime
def prime(x):
    ''' Returns the xth prime '''
    lastn = prime_list[-1]
    while len(prime_list) <= x:                 # Keep working until
    we've got the xth prime
        lastn = lastn + 1                        # Check the next
    number
```

[Property]
category: class implementation
language: Python
difficulty: medium
concepts: prime number memoization, n-th prime calculation, efficiency
    in prime checks

[Instruction]
Implement a Python class with a method `find_nth_prime_memoized(n)`
    that returns the nth prime number, where n is a positive integer.
    Utilize memoization to store already found prime numbers in a list
     within the class to avoid recalculating primes for multiple
    method calls. Ensure the method efficiently updates and references
     this list to find new primes as needed. Additionally, enhance
    performance by avoiding unnecessary prime checks.

[Response]
Implementing the required Python class with the method `
    find_nth_prime_memoized(n)` involves several key steps:

1. The class should have a constructor `__init__` that initializes the
     cache list to store the memoized prime numbers. Initially, the
    list can be seeded with the first two prime numbers, 2 and 3.
2. The method `find_nth_prime_memoized` will check if the requested `n
    `th prime is already in the memoized list. If it is, it will
    return that prime directly. If not, it will start from the last
    cached prime and continue to find and cache new primes until the `
    n`th prime is found.
3. A helper method `is_prime` can be defined to check if a number is
    prime. To efficiently check for primality:
    - It checks if the number is divisible by each of the primes
    already in the cache.
    - It only checks up to the square root of the number to be tested,
     implemented by comparing the square of the prime with the number.
    - It skips even numbers by incrementing by 2 for each prime check.
     This is sound since the cache is initialized with the first **two
    ** primes, 2 and 3.

Here is how you could implement such a class:

```python
class PrimeFinder:
    def __init__(self):
        self.primes = [2, 3]

    def is_prime(self, num: int) -> bool:
        if num < 2:
            return False
        for prime in self.primes:
            if prime * prime > num:
                break
            if num % prime == 0:
                return False
        return True
```

```python
    def find_nth_prime_memoized(self, n: int) -> int:
        if n <= len(self.primes):
            return self.primes[n - 1]

        current_num = self.primes[-1] + 2
        while len(self.primes) < n:
            if self.is_prime(current_num):
                self.primes.append(current_num)
            current_num += 2

        return self.primes[n - 1]
```

[Tests]
Here's how you can test this class by verifying that the class
    correctly finds the first few primes and that subsequent calls
    return the memoized results directly:

```python
first_few_primes = [2, 3, 5, 7, 11, 13, 17, 19, 23, 29]
prime_finder = PrimeFinder()
for i, prime in enumerate(first_few_primes, 1):
    assert prime_finder.find_nth_prime_memoized(i) == prime

for i, prime in enumerate(first_few_primes, 1):
    assert prime_finder.find_nth_prime_memoized(i) == prime
```

### Example 4
[Code]
```
    return (a + 1) * (b + 1) - 1
def keys_count(a, b):
    return powerset(a, b) * 2 - a - b
def formula(k):
    if k % 2 != 0:
        return ((k + 1) ** 2) / 2 + k + 1
    else:
        return (k ** 2) / 2 + 2 * k + 1
def multiset_powerset(multiset):
    n = len(multiset)
    c = [0] * n
    while True:
        changed = False
        i = n - 1
        while i >= 0 and not changed:
```

[Property]
category: function implementation
language: Python
difficulty: easy
concepts: special formula implementation, odd and even number handling
    , function composition

[Instruction]
Here are two special formulas:

$$
f_1(a, b) = (a + 1) \cdot (b + 1) - 1
$$

$$
f_2(k) = \begin{cases}
    \frac{(k + 1)^2}{2} + k + 1 & \text{if } k \text{ is odd} \\
    \frac{k^2}{2} + 2k + 1 & \text{if } k \text{ is even}
\end{cases}
$$

```
\end{cases}
$$

Write a Python function to return $f_2(f_1(a, b))$ for given `a` and `
    b`.

[Response]
Based on the formulas you provided, we can define two Python functions
    , `f1(a, b)` and `f2(k)`, respectively, and then combine them to
    calculate $f2(f1(a, b))$ for given `a` and `b`.

Here is the implementation:

```python
def f1(a, b):
    return (a + 1) * (b + 1) - 1

def f2(k):
    if k % 2 == 0:
        return (k**2) / 2 + 2 * k + 1
    else:
        return ((k + 1)**2) / 2 + k + 1

def f1_f2_combined(a, b):
    return f2(f1(a, b))
```

In this implementation:

- `f1(a, b)` calculates the first formula you've given.
- `f2(k)` calculates the second formula, with a conditional branch for
    whether `k` is even or odd.
- `f1_f2_combined(a, b)` uses the output of `f1(a, b)` as the input
    for `f2(k)` and returns the result.

[Tests]
To test this combined function, you can simply compare it with the
    direct application of `f2` to the result of `f1` for different
    values of `a` and `b`:

```python
assert f1_f2_combined(2, 3) == f2(f1(2, 3))
assert f1_f2_combined(4, 5) == f2(f1(4, 5))
assert f1_f2_combined(6, 7) == f2(f1(6, 7))
```

### Example 5
[Code]
    while left<len(arr) and ryt >= start and left <= ryt:
        mid = (left+ryt)//2
        if arr[mid] == target:
            return mid
        elif arr[mid] > target:
            ryt = mid-1
        else:
            left = mid+1
    return left
def tripletsSorting(nums, t):
    # TimeComplexity = O((n^2)logn)
    nums.sort()
    count = 0
    for i in range(len(nums)):

[Property]
category: function implementation
```

```
language: Python
difficulty: easy
concepts: insertion point for a sorted array, optimized time
    complexity
```

[Instruction]
Design a Python function that takes a sorted array and a target value,
    and return a valid index where target can be inserted to maintain
    the array's sorted order. Optimize the function to run in
    logarithmic time complexity.

For example, given `array = [1, 3, 5, 5, 6]` and `target = 5`, the
    function should return either 2 or 3 because 5 presents at both
    indices 2 and 3.

[Response]
To solve this problem efficiently and ensure logarithmic time
    complexity, we can use a binary search algorithm. Compared with a
    standard binary search that looks for an exact match, we can
    modify the algorithm such that when the target is not found, we
    return the `left` bound, which represents the index where the
    target can be inserted to maintain the array's sorted order. Since
    any valid index for insertion is acceptable, we can direclty
    return the index if there is an exact match.

Here is a Python function that implements this approach:

```python
from typing import List

def search_insert_position(nums: List[int], target: int) -> int:
    """
    Finds the index to insert `target` into `nums` to maintain sorted
    order. For example,
    given `nums = [1, 3, 5, 5, 6]` and `target = 5`, this function
    returns either 2 or 3,
    as `target` can be inserted at index 2 or 3 to maintain the sorted
     order.
    """
    left, right = 0, len(nums) - 1
    while left <= right:
        mid = (left + right) // 2
        if nums[mid] == target:
            return mid
        elif nums[mid] < target:
            left = mid + 1
        else:
            right = mid - 1
    return left
```

This implementation ensures that when `target` is not found, `left`
    represents the correct insertion index. This is because at the end
     of the loop, all elements to the left of `left` are smaller than
    `target`, all elements to the right of `left` are equal to or
    larger than `target`, and `left > right`. Therefore, `left` is the
     correct index for insertion to maintain the sorted order.

[Tests]
You can test the function with scenarios where the target is found at
    multiple indices and where the target is not present in the array:

```python
assert search_insert_position([1, 3, 5, 5, 6], 5) in [2, 3]
assert search_insert_position([1, 3, 5, 5, 6], 2) == 1
```

```
assert search_insert_position([1, 3, 5, 5, 6], 7) == 5
assert search_insert_position([1, 3, 5, 5, 6], 0) == 0
```

### Example 6
[Code]
```
def decompress(self):
    source = self.compressed
    if isinstance(source, (bytes, bytearray)):
        return self.decompress_bytes()
    pos = 0
    node = self.root
    res = bytearray()

    while pos < len(source):
        code = int(source[pos])
        child = node.children[code]
        if child.is_leaf:
            res.append(child)
            node = self.root
        else:
            node = child
        pos += 1

    return bytes(res)
```

[Property]
category: function implementation
language: Python
difficulty: medium
concepts: decoding compressed data, binary tree traversal, byte array
    processing

[Instruction]
Write a Python function `huffman_decompress` that takes two arguments:
    `compressed_data`, a byte array of 0/1 sequence representing
    Huffman compressed data, and `huffman_tree`, a Huffman tree that
    stores the actual data (each leaf is associated with a character).
    The function should return the original uncompressed data as `
    bytes`. Define the `HuffmanNode` class yourself as a general tree
    structure with a `children` list.

[Response]
To accomplish the Huffman decompression, we first need to define the `
    HuffmanNode` class, which, as required, should be a general tree
    structure with a `children` list and an optional `value` attribute
    to store the character associated with a leaf node.

Then, we'll implement the `huffman_decompress` function. This function
    will traverse the Huffman tree according to the bits in `
    compressed_data`. When it encounters a leaf node, the associated
    character will be appended to the result, and the traversal
    restarts from the root of the tree. This process continues until
    the entire `compressed_data` is processed.

You said that `compressed_data` is a byte array of 0/1 sequence, so I
    assume that each byte, not each bit, represents a step in the
    Huffman tree. In practice, it is more efficient

Here's how we can implement this:

```python
class HuffmanNode:
    def __init__(self, value=None):
        self.value = value
```

```
        self.children = []

def huffman_decompress(compressed_data, huffman_tree):
    result = []
    node = huffman_tree

    # Each byte represents a bit in `compressed_data`
    for bit in compressed_data:
        if len(node.children) > 0:
            node = node.children[int(bit)]
        if len(node.children) == 0:
            result.append(node.value)
            node = huffman_tree

    return "".join(result).encode()
```

In this implementation, each byte in `compressed_data` represents a
    bit, guiding the traversal of the Huffman tree. Characters are
    appended upon reaching leaf nodes and the result is then encoded
    to `bytes`.

[Tests]
You can test this function with a sample Huffman tree and compressed
    data to verify that the decompression works correctly:

```python
root = HuffmanNode()
root.children = [HuffmanNode('A'), HuffmanNode('B')]
root.children[0].children = [HuffmanNode('C'), HuffmanNode('D')]
root.children[1].children = [HuffmanNode('E'), HuffmanNode('F')]
compressed_data = bytearray([0, 1, 0, 0, 1, 1])
# 01 -> D, 00 -> C, 11 -> F
assert huffman_decompress(compressed_data, root) == b'DCF'
```

### Example 7
[Code]
def format_size(num):
    """http://stackoverflow.com/a/1094933
    """
    for x in ['bytes', 'KB', 'MB', 'GB']:
        if num < 1024.0 and num > -1024.0:
            return "%3.1f%s" % (num, x)
        num /= 1024.0
    return "%3.1f%s" % (num, 'TB')
assert format_size(1024**2 - 1) == '1024.0KB'
assert format_size(1024*512) == '512.0KB'

[Property]
category: function implementation
language: Python
difficulty: easy
concepts: arithmetic operations for size conversion, rounding numbers,
    dynamic unit selection, string interpolation

[Instruction]
Create a Python function to format a file size in bytes into a human-
    readable string representation, using 'bytes', 'KB' (kilobytes), '
    MB' (megabytes), 'GB' (gigabytes), or 'TB' (terabytes) units. The
    output should be rounded to one decimal place and follow the
    format: "{X.Y}{Unit}", where "Unit" is the appropriate size unit
    and "X.Y" is the rounded size. For example, `format_size(1024 *
    512)` should return `'512.0KB'`, and `format_size(1024**2 - 1)`
    should return `'1024.0KB'`.
```

[Response]
To create such a function, we can use a loop to iterate through a
    predefined list of units (`['bytes', 'KB', 'MB', 'GB', 'TB']`) and
     check if the given number of bytes fits into the current unit by
    comparing it with 1024.0 and -1024.0 (for handling negative sizes)
    . We'll keep dividing the number by 1024.0 until the correct unit
    is found where the number is less than 1024.0. Then, we'll format
    the result using the appropriate unit and the rounded size.

```python
def format_size(size_bytes):
    units = ['bytes', 'KB', 'MB', 'GB', 'TB']

    # Directly return for 0 bytes to avoid division by zero in loop
    if size_bytes == 0:
        return "0.0bytes"

    num = abs(size_bytes)
    unit = units[0]
    for u in units[1:]:
        if num < 1024.0:
            break
        num /= 1024.0
        unit = u
    formatted_size = f"{num:.1f}{unit}"
    return formatted_size if size_bytes >= 0 else f"-{formatted_size}"
```

This implementation handles both positive and negative file sizes and
    returns the formatted size as a string, even though negative sizes
     are not common in real-world scenarios.

[Tests]
We can write tests that cover both positive and negative sizes across
    different units to validate the correctness of the `format_size`
    function:

```python
test_data = [
    (1024 ** 2, "1.0MB"),
    (1024 ** 2 - 1, "1024.0KB"),
    (1024 ** 3 * 512, "512.0GB"),
    (1024 ** 4 * 512, "512.0TB"),
]
for size, expected_output in test_data:
    assert format_size(size) == expected_output
    assert format_size(-size) == f"-{expected_output}"
```

### Example 8
[Code]
def unify_stringlist(L: list):
    """ Adds asterisks to strings that appear multiple times, so the
    resulting
    list has only unique strings but still the same length, order, and
     meaning.
    For example:
        unify_stringlist(['a','a','b','a','c']) -> ['a','a*','b','a
    **','c']
    """
    assert(all([isinstance(l,str) for l in L]))
    return [L[i]+"*"*L[:i].count(L[i]) for i in range(len(L))]
assert unify_stringlist(list("abc")) == list("abc")
```

```
[Property]
category: function implementation
language: Python
difficulty: easy
concepts: list comprehension, unique string creation, precondition
    checking for type validation

[Instruction]
Design a Python function to encode a list of strings into a unique
    list with the same length, order, and meaning. The new list is
    formed by appending asterisks (*) to duplicate strings. For
    instance, given ['a', 'a', 'b', 'a', 'c'], the function should
    return ['a', 'a*', 'b', 'a**', 'c'], while for ['a', 'b', 'c'] it
    should return ['a', 'b', 'c'] without any changes. Write
    assertions to ensure the input is a list of strings.

Solve the problem in two lines of code, one for the precondition check
     and the other for the main logic.

[Response]
You can achieve this functionality by first ensuring the input meets
    your requirements using an `assert` statement with list
    comprehension. Then, use another list comprehension to iterate
    through the input list and append `'*'` to the strings. The number
     of `'*'` is the number of occurrences of the string before the
    current index.

Here's how you can do it in two lines of Python code:
```

```python
def encode_strings(input_list):
    assert all(isinstance(item, str) for item in input_list), "Input
    must be a list of strings"
    return [s + '*' * input_list[:i].count(s) for i, s in enumerate(
    input_list)]
```

```
Note that although the function is concise, it is not the most
    efficient solution for large lists, as the `count` method has a
    time complexity of O(n) for each iteration.

[Tests]
You can test the function with the provided examples to ensure it
    correctly encodes the strings as expected:
```

```python
assert encode_strings(['a', 'a', 'b', 'a', 'c']) == ['a', 'a*', 'b', '
    a**', 'c']
assert encode_strings(['a', 'b', 'c']) == ['a', 'b', 'c']
```

```
### Example 9
[Code]
def _create_folds_list(data, count):
    """
    Creates folds from the given data.

    :param data: the data to fold
    :param count: the number of folds to create

    :return: a list of folds
    """

    fold_count = len(data) / count
    folds = list()
```

```
    for fold_index in range(count):
        low = int(fold_index * fold_count)
        high = int((fold_index + 1) * fold_count)

        fold = data[low:high]
        folds.append(fold)

    return folds
```

[Property]
category: function implementation
language: Python
difficulty: medium
concepts: data partitioning and slicing, learning from input-output
    examples

[Instruction]
Construct a Python function `create_folds(data, count)` to create a
    list of folds from the given data, where each fold is a
    subsequence of the original data. The length of each fold should
    be approximately equal with at most a difference of 1. The
    function should satisfy the following assertions:

```python
assert create_folds(range(12), 2) == [range(0, 6), range(6, 12)]
assert create_folds(range(8), 2) == [range(0, 4), range(4, 8)]
assert create_folds(range(25), -5) == []
assert create_folds(range(6), 2) == [range(0, 3), range(3, 6)]
```

[Response]
To create a Python function that generates a list of folds from given
    data with each fold having approximately equal length (with at
    most a difference of 1 between any two folds), you can follow
    these steps:

1. Return an empty list immediately if `count` is non-positive since
   it's not possible to create a valid number of folds.
2. Divide the total length of the data by `count`, rounding down, to
   get the minimum number of items each fold should have.
3. Calculate the remainder to see how many extra items need to be
   distributed among the folds. They are distributed one by one to
   the first `remainder` folds.
4. Use a loop to create each fold, adjusting the starting and ending
   indices based on the calculations above to ensure each fold has
   the correct number of items.

Here's how you can implement this:

```python
def create_folds(data, count):
    # Return an empty list if count is non-positive
    if count <= 0:
        return []

    data_length = len(data)
    fold_size = data_length // count
    remainder = data_length % count

    folds = []
    start_index = 0
    for i in range(count):
        # Add 1 element to each of the first `remainder` folds
```

```
            end_index = start_index + fold_size + (1 if i < remainder else
    0)
            folds.append(range(start_index, end_index))
            start_index = end_index
    return folds
```

This implementation ensures that each fold has an equal length with at
    most a difference of 1 between any two folds. It handles edge
    cases, such as a non-positive `count`, by returning an empty list
    as specified.

[Tests]
You can test your code with the provided assertions to verify that the
    function works correctly:

```python
assert create_folds(range(12), 2) == [range(0, 6), range(6, 12)]
assert create_folds(range(8), 2) == [range(0, 4), range(4, 8)]
assert create_folds(range(25), -5) == []
assert create_folds(range(6), 2) == [range(0, 3), range(3, 6)]
```

### Example 10
[Code]
def duration(string):
    match = re.match(PATTERN, string)
    if not match:
        raise ValueError('Invalid duration: {}'.format(string))

    suffix = match.group(2)
    if suffix not in SUFFIX_MAP:
        raise ValueError('Invalid duration suffix: {}'.format(string))

    return int(match.group(1)) * SUFFIX_MAP[suffix]

[Property]
category: function implementation
language: Python
difficulty: medium
concepts: regular expression for pattern matching, dictionary lookup
    for suffix mapping, error handling with exceptions

[Instruction]
Create a Python function that converts a string representation of a
    time duration into its total number of seconds.
The input string should follow a format that includes a numeric part
    followed by a time unit suffix without any spaces between.
The time unit can be seconds in the form of 's' or 'sec', minutes in
    the form of 'm' or 'min', hours in the form of 'h' or 'hour', and
    days in the form of 'd' or 'day' (units are in small case).
If the input format is invalid, the function should raise a ValueError
    .

For example, the function should pass the following assertions:

```python
assert parse_duration('15s') == 15
assert parse_duration('2min') == 120
assert parse_duration('1h') == 3600
assert parse_duration('1d') == 86400
try:
    parse_duration('30')
except ValueError as e:
    pass
```
```

```
```

[Response]
The desired function aims to transform a string representation into
    its total number of seconds. The implementation can be split into
    two parts:

Input validation:

1. The input string includes two parts: a number as the prefix and the
    unit as the suffix.
2. We can use regular expression to assert the prefix must be a number
    (d+) and suffix must be one of s/sec/m/min/h/hour/d/day.
3. If the input string cannot the regular expression, we raise a
    ValueError.

Result computation:

1. Based on the matches of regular expression, we capture the number
    and unit part.
2. We convert the unit to seconds: 's'/'sec' -> 1 second, 'm'/'min' ->
    60 seconds, 'h'/'hour' -> 3600 seconds, 'd'/'day' -> 86400
    seconds.
3. We multiply the seconds of the unit and the prefix number and
    return it as an integer.

Here's how you can implement this:

```python
import re

def parse_duration(duration_str):
    pattern = re.compile(r"(\d+)(s|sec|m|min|h|hour|d|day)$")
    match = pattern.match(duration_str)
    if not match:
        raise ValueError(f"Invalid duration: {duration_str}")
    value, suffix = match.groups()
    value = int(value)
    suffix_map = {
        's': 1, 'sec': 1,
        'm': 60, 'min': 60,
        'h': 3600, 'hour': 3600,
        'd': 86400, 'day': 86400
    }
    return value * suffix_map[suffix]
```

[Tests]
You can test your code on various valid and invalid inputs to ensure
    that the function behaves as expected:

```python
# Test additional valid inputs with different suffix representations
assert parse_duration('10sec') == 10, "Check for 'sec' suffix"
assert parse_duration('3min') == 180, "Check for 'min' suffix"
assert parse_duration('5h') == 18000, "Check for 'h' suffix"
assert parse_duration('2hour') == 7200, "Check for 'hour' suffix"
assert parse_duration('1day') == 86400, "Check for 'day' suffix"

# Test edge cases
assert parse_duration('0s') == 0, "Zero duration"
assert parse_duration('0min') == 0, "Zero minutes"

# Test inputs with leading zeros
assert parse_duration('001s') == 1, "Leading zeros"
```

```
assert parse_duration('010min') == 600, "Leading zeros with minutes"

# Test case sensitivity and invalid suffix handling
def assert_value_error(duration_str):
    try:
        parse_duration(duration_str)
    except ValueError:
        pass

assert_value_error('10S')
assert_value_error('2hours')
assert_value_error('5 mins')
assert_value_error('h1')
assert_value_error('100')

# Test extremely large numbers
assert parse_duration('1000000s') == 1000000, "Very large number of
    seconds"
```

### Example 11
[Code]
```
class Restaurant():

    def __init__(self, name, cuisine_type):
        self.name = name.title()
        self.cuisine_type = cuisine_type
        self.number_served = 0

    def describe_restaurant(self):
        msg = f"{self.name} ... {self.cuisine_type}."
        print(f"\n{msg}")

    def open_restaurant(self):
        msg = f"{self.name} ... \n"
        print(f"\n{msg}")

    def set_number_served(self, number_served):
        self.number_served = number_served # ...

    def increment_number_served(self, additional_served):
        self.number_served += additional_served # ...
```

[Property]
category: class implementation
language: Python
difficulty: hard
concepts: object-oriented programming, string formatting and
    manipulation, encapsulation, input validation

[Instruction]
Develop a Python class `StringEnhancer` that offers advanced tools for
    managing a string. The class should maintain an internal state
    that is initially an empty string. The class should support
    extending the string with additional prefix or suffix, converting
    the entire string to lowercase or uppercase, and altering specific
     characters within the string. All the operations are communicated
     through a central method: `apply_operation(self, operation: str)`
    .

The method `apply_operation` interprets the operations specified in
    the form of `"[opname]:[arg]"`. Here are the detailed formats for
    each operation, explained through examples:
- Prefix addition: `apply_operation("prefix:Hello, ")`
- Suffix addition: `apply_operation("suffix: World!")`

- Lowercase conversion: `apply_operation("lower:")`
- Uppercase conversion: `apply_operation("upper:")`
- Character replacement: `apply_operation("replace:a->e")`

The class should handle erroneous inputs gracefully, such as incorrect
      operation formats or arguments. Additionally, implement the `
    __str__` method to return a formatted string showing the current
    value of the string and a list of operations that have been
    applied to it.

[Response]
The goal is to create a Python class `StringEnhancer` capable of
    managing a string through various operations, using a central
    method `apply_operation`. The class also needs to handle errors
    gracefully and record the operation history. Here are the key
    steps to implement this class:

- Initialize the class with an empty string and an empty list to store
      the operations.
- Inside the `apply_operation` method:
    - Parse the operation string to determine the operation type and
    argument (if any).
    - Gracefully handle any errors in the input format by printing an
    error message without actually raising an exception.
    - Execute the correct string manipulation based on the operation.
    - Append the operation to the records list.
- Implement the `__str__` method by concatenating the current string
    value and the recorded operations.

Here's how you can implement the `StringEnhancer` class:

```python
class StringEnhancer:
    def __init__(self):
        self.string = ""
        self.operations = []

    def apply_operation(self, operation: str):
        parts = operation.split(":")
        if len(parts) == 1:
            print("Invalid operation format. Expected '[opname]:[arg
    ]'")
            return

        opname, arg = parts[0], parts[1]
        if opname == "prefix":
            self.string = arg + self.string
        elif opname == "suffix":
            self.string += arg
        elif opname == "lower":
            self.string = self.string.lower()
        elif opname == "upper":
            self.string = self.string.upper()
        elif opname == "replace":
            if "->" not in arg:
                print("Invalid replace operation format. Expected '
    replace:a->b'")
                return
            old_char, new_char = arg.split("->")
            self.string = self.string.replace(old_char, new_char)
        else:
            print(f"Unknown operation: {opname}")

        self.operations.append(operation)
```

```python
    def __str__(self):
        operation_history = ', '.join(self.operations)
        return f"Current String: {self.string}\nOperations: {
    operation_history}"
```

[Tests]
To test the class, you should apply different operations and check if
    the string and operation history are updated correctly:

```python
enhancer = StringEnhancer()
enhancer.apply_operation("prefix:Hello")
enhancer.apply_operation("suffix:, World!")
enhancer.apply_operation("lower:")
enhancer.apply_operation("replace:o->a")

expected_operation_history = ["prefix:Hello", "suffix:, World!", "
    lower:", "replace:o->a"]
expected_operation_history_str = ', '.join(expected_operation_history)
expected_str_value = "hella, warld!"
assert enhancer.operations == expected_operation_history
assert str(enhancer) == f"Current String: {expected_str_value}\
    nOperations: {expected_operation_history_str}"
```

### Example 12
[Code]
```
while cur_num < len(sorted_importances):
    cluster_idx = int(sorted_importances[cur_num][1])
    filter_idx = int(sorted_importances[cur_num][2])

    if tmp_pruning_quotas[cluster_idx] > 0:
        tmp_pruning_quotas[cluster_idx] -= 1
    else:
        cur_num += 1
        continue

    cluster = self.pruned_module_groups_info.get_cluster_by_id(
    cluster_idx)
    for node in cluster.elements:
```

[Property]
category: function implementation
language: Python
difficulty: easy
concepts: iteration with a while loop, index-based data access,
    conditional branching, nested loops

[Instruction]
Create a Python function that identifies all pairs of elements within
    a list of integers where the second element of the pair is at
    least greater than the first by `growth_ratio` and the indices of
    the elements are within a specified `maximum_distance` from each
    other.

[Response]
You can write a Python function `find_pairs` that takes three
    parameters:

- `numbers`: The list of integers.
- `growth_ratio`: A non-negative `float` that specifies the ratio by
    which the second element should be greater than the first.

- `maximum_distance`: A non-negative integer that indicates the
    maximum index distance allowed between the two elements in the
    pair.

The function will return a list of tuples where each tuple contains
    the indices of the pair of elements that satisfy the conditions.

Here is how you can implement this function:

```python
from typing import List, Tuple

def find_pairs(numbers: List[int], growth_ratio: float,
    maximum_distance: int) -> List[Tuple[int, int]]:
    """
    Find pairs of elements within a list of integers where the second
    element is at least `growth_ratio` greater than the first and the
    indices are within `maximum_distance` from each other.
    """
    pairs: List[int] = []
    for i in range(len(numbers)):
        # For each number, look ahead up to `maximum_distance`
    elements
        for j in range(i + 1, min(i + 1 + maximum_distance, len(
    numbers))):
            if numbers[j] - numbers[i] >= growth_ratio:
                pairs.append((i, j))
    return pairs
```

This function iterates over the pairs of elements in the list whose
    indices satisfy the distance constraint and stores the valid pairs
     that meet the growth ratio condition.

[Tests]
To ensure the function's correct behavior, you can compare your
    implementation against a brute-force approach that leverages list
    comprehension to first generate all possible pairs and then filter
     them based on the conditions. The tests should cover both the
    basic functionality where pairs are found, and edge cases where no
     pairs are found, while also altering the `growth_ratio` and `
    maximum_distance` parameters to cover more cases:

```python
def brute_force_pairs(numbers: List[int], growth_ratio: float,
    maximum_distance: int) -> List[Tuple[int, int]]:
    return [
        (i, j)
        for i in range(len(numbers))
        for j in range(i + 1, min(i + 1 + maximum_distance, len(
    numbers)))
        if numbers[j] - numbers[i] >= growth_ratio
    ]

test_cases = [
    ([30, 15, 60, 7, 77, 13, 10], 8, 3),
    ([30, 15, 60, 7, 77, 13, 10], 0.2, 3),
    ([30, 15, 60, 7, 77, 13, 10], 30, 0),
    ([30, 15, 60, 7, 77, 13, 10], 0, 2),
    ([], 1, 3),
    ([8, 8, 8, 8], 1, 0)
]

for numbers, growth_ratio, maximum_distance in test_cases:
```

```
    expected_pairs = brute_force_pairs(numbers, growth_ratio,
    maximum_distance)
    assert find_pairs(numbers, growth_ratio, maximum_distance) ==
    expected_pairs
```

### Example 13
[Code]
```python
def accuracy(output, target, topk=(1,)):
    """Computes the precision@k for the specified values of k"""
    maxk = max(topk)
    batch_size = target.size(0)

    _, pred = output.topk(maxk, 1, True, True)
    pred = pred.t()
    correct = pred.eq(target.view(1, -1).expand_as(pred))

    res = []
    for k in topk:
        correct_k = correct[:k].view(-1).float().sum(0)
        res.append(correct_k.mul_(100.0 / batch_size))
    return res
```

[Property]
category: function implementation
language: Python
difficulty: medium
concepts: top-k accuracy computation, tensor operations in PyTorch,
    batch processing, default arguments

[Instruction]
Create a Python function `calculate_top_k_accuracy` using PyTorch that
    calculates the top-k accuracy from a batch of predictions and
    actual labels. It should accept two tensors, `predictions` and `
    targets`, and an integer `k` (default 3). The function should
    return the average top-k accuracy as a float, determining if the
    true label is within the top `k` predictions for each input.

[Response]
The function will take the following parameters as input:

- `predictions`: a tensor of shape `(batch_size, num_classes)` that
    contains the predicted scores for each class.
- `targets`: a tensor of shape `(batch_size,)` that contains the true
    class labels for each input.
- `k`: an integer representing the number of top predictions to
    consider for computing the accuracy. The default value should be
    3.

Here is how you can write this function using PyTorch:

```python
import torch

def calculate_top_k_accuracy(predictions: torch.Tensor, targets: torch
    .Tensor, k: int = 3) -> float:
    # Ensure this function works even if k is greater than the number
    of classes
    k = min(k, predictions.size(1))
    _, top_k_indices = predictions.topk(k, dim=1)
    target_expanded = targets.unsqueeze(1).expand_as(top_k_indices)
    correct_predictions = top_k_indices.eq(target_expanded)
    correct = correct_predictions.any(dim=1)
    accuracy = correct.float().mean()
    return accuracy.item()
```

```
```

The function retrieves the top k predictions' indices, aligns shapes
    for comparison with true labels, computes accuracy by checking
    matches, and returns the average accuracy as a float.

[Tests]
```python
# Test 1: Basic functionality test
k = 1
labels = torch.tensor([2, 0, 1, 1])
preds = torch.tensor([
    [0.1, 0.2, 0.9], # yes
    [1.0, 0.2, 0.3], # yes
    [0.3, 1.1, 0.2], # yes
    [0.6, 0.1, 0.3], # no
])
accuracy = calculate_top_k_accuracy(preds, labels, k)
assert accuracy == 0.75

# Test 2: Edge case with k = number of classes
k = 3
preds = torch.tensor([
    [0.5, -1.2, 0.3], # yes
    [-1.0, 0.1, 1.2], # yes
])
labels = torch.tensor([0, 2])
accuracy = calculate_top_k_accuracy(preds, labels, k)
assert accuracy == 1.0

# Test 3: k greater than the number of classes should still work
preds = torch.tensor([
    [0.1, -0.2], # yes
    [-0.3, 0.4], # yes
])
labels = torch.tensor([1, 0])
accuracy = calculate_top_k_accuracy(preds, labels, k=5)
assert accuracy == 1.0
```

### Example 14
[Code]
def alternatingCharacters(s):
    i = 0
    j = 1
    count = 0
    while j<len(s):
        if s[j] == s[i]:
            while j<len(s) and s[j] == s[i]:
                j+=1
            count += j-i-1
        i = j
        j += 1
    return count

[Property]
category: function implementation
language: Python
difficulty: easy
concepts: 2-pointer technique, conditional loop, string traversal and
    character comparison, count accumulation

[Instruction]
Write a Python function called `count_conditional_matches` that takes
    two strings, `str1` and `str2`, as well as a boolean flag `strict`

as arguments. The function should count how many times characters from the end of `str1` match with characters from the beginning of `str2` when both strings are traversed simultaneously. The traversal should continue until a pair of characters do not match or until a specific condition influenced by the `strict` flag is met:

- If `strict` is `True`, stop counting as soon as a non-matching pair is found.
- If `strict` is `False`, skip the non-matching character in `str1` and continue matching with the next character.

The function should return the total count of matching characters. This implementation requires managing two pointers for the characters in `str1` and `str2`, respectively, and using a conditional loop to handle different stopping criteria based on the `strict` parameter.

[Response]
We can use two pointers, one for each string, to iterate over the characters from the end of str1 and the beginning of str2. The loop continues either until the characters do not match (for both strict and non-strict cases) or until it exhausts one of the strings. The `strict` flag controls the behavior when a mismatch is encountered.

```python
def count_conditional_matches(str1, str2, strict):
    # Initialize pointers for the end of str1 and the start of str2
    i, j = len(str1) - 1, 0
    count = 0

    # Loop as long as i is non-negative and j is less than the length
    of str2
    while i >= 0 and j < len(str2):
        if str1[i] == str2[j]:
            count += 1
            i -= 1
            j += 1
        elif strict:
            break
        else:
            i -= 1
    return count
```

[Tests]
The function can be tested with various inputs to ensure it handles both strict and non-strict cases correctly:

```python
# Test function with matching characters
assert count_conditional_matches("hello", "o", True) == 1
assert count_conditional_matches("fancy", "nancy", False) == 2
assert count_conditional_matches("abcde", "e", False) == 1
assert count_conditional_matches("helicopter", "ranges", True) == 1

# Test function with no matches
assert count_conditional_matches("apple", "banana", True) == 0
assert count_conditional_matches("abcdef", "xyzab", False) == 0

# Test when both strings are empty
assert count_conditional_matches("", "", True) == 0
assert count_conditional_matches("", "", False) == 0
```

```
# Test when one string is empty
assert count_conditional_matches("nonempty", "", True) == 0
assert count_conditional_matches("", "nonempty", False) == 0
```

### Example 15
[Code]
```
def oxford_comma_text_to_list(phrase):
    """Examples:
    - 'Eeeny, Meeny, Miney, and Moe' --> ['Eeeny', 'Meeny', 'Miney', '
    Moe']
    - 'Black and White' --> ['Black', 'White']
    - 'San Francisco and Saint Francis' -->
        ['San Francisco', 'Saint Francisco']
    """
    items = []
    for subphrase in phrase.split(', '):
        items.extend(
            [item.strip() for item in subphrase.split(' and ')])
    return items
```

[Property]
category: function implementation
language: Python
difficulty: easy
concepts: string manipulation and parsing, list comprehension,
    iterative list extension, handling text connectors

[Instruction]
Create a concise Python function that transforms a string into a list
    of individual items. The input string contains a list of items
    separated by commas and the last item preceded by an optional "and
    ". For example:
- Input: "Eeeny, Meeny, Miney, Moe" --> Output: ['Eeeny', 'Meeny', '
    Miney', 'Moe']
- Input: "Eeeny, Meeny, Miney, and Moe" --> Output: ['Eeeny', 'Meeny',
    'Miney', 'Moe']
- Input: "San Francisco and San Mateo" --> Output: ['San Francisco', '
    San Mateo']
Remove any leading or trailing whitespace.

[Response]
Here's a concise function `string_to_list` to transform a comma-
    separated string into a list, replacing " and " with a comma,
    splitting the result, and stripping whitespace:

```python
def string_to_list(s):
    return [item.strip() for item in s.replace(" and ", ",").split
    (",") if item.strip() != ""]
```

This approach ensures no empty strings are included if there are
    excess commas or spaces.

[Tests]
We can test this function on different input strings (e.g., with and
    without the "and" connector, different numbers of items):

```python
assert string_to_list("Eeeny, Meeny, Miney, Moe") == ['Eeeny', 'Meeny
    ', 'Miney', 'Moe']
assert string_to_list("Eeeny, Meeny, Miney, and Moe") == ['Eeeny', '
    Meeny', 'Miney', 'Moe']
```

```
assert string_to_list("San Francisco and San Mateo") == ['San
    Francisco', 'San Mateo']
assert string_to_list("Single Item") == ['Single Item']
```

### Example 16
[Code]
def get_hashes(buf):
    hexdigests = namedtuple('Digests', 'md5 sha1 sha256 sha512 ctph')
    if isinstance(buf, str):
        buf = open(buf, 'rb').read()
    md5 = hashlib.md5(buf).hexdigest()
    sha1 = hashlib.sha1(buf).hexdigest()
    sha256 = hashlib.sha256(buf).hexdigest()
    sha512 = hashlib.sha512(buf).hexdigest()
    ctph = ssdeep.hash(buf)
    return hexdigests._make((md5, sha1, sha256, sha512, ctph))

[Property]
category: function implementation
language: Python
difficulty: easy
concepts: binary string hash computation, file content reading, string
    hashing, named tuples, conditional file content reading

[Instruction]
Implement a Python function that computes different types of hash
    values for the binary content of a file whose path is defined by
    an input string. You should calculate 'md5', 'sha1', 'sha256', and
     'sha512' and return a `namedtuple` of these fields. When the
    input string does not correspond to a file, the function should
    directly compute the hash values of path itself.

[Response]
To compute the hashes of a file or a string, we'll follow these steps:

1. If the input is a valid file path, read the binary content of the
    file.
2. Otherwise, convert the input string to a binary string.
3. Compute hash values of all types using the `hashlib` module.
4. Return the hash values as a `namedtuple` for easy access.

Here's how you can implement this function:

```python
import hashlib
from collections import namedtuple
from pathlib import Path

HashValues = namedtuple('HashValues', ['md5', 'sha1', 'sha256', '
    sha512'])

def compute_hashes(file_path):
    try:
        with open(file_path, 'rb') as f:
            content = f.read()
    # Handle cases where the file does not exist or the path is a
    directory
    except (FileNotFoundError, IsADirectoryError):
        content = file_path.encode()

    md5_hash = hashlib.md5(content).hexdigest()
    sha1_hash = hashlib.sha1(content).hexdigest()
    sha256_hash = hashlib.sha256(content).hexdigest()
    sha512_hash = hashlib.sha512(content).hexdigest()
```

```
        return HashValues ( md5_hash , sha1_hash , sha256_hash , sha512_hash )
```

[Tests]
To test the function , you can provide string inputs to avoid file I/O
    operations and verify that the hash values are correctly computed:

```python
string = 'random string '
binary = string . encode ()
computed_hashes = compute_hashes ( string )
assert computed_hashes . md5 == hashlib . md5 ( binary ) . hexdigest ()
assert computed_hashes . sha1 == hashlib . sha1 ( binary ) . hexdigest ()
assert computed_hashes . sha256 == hashlib . sha256 ( binary ) . hexdigest ()
assert computed_hashes . sha512 == hashlib . sha512 ( binary ) . hexdigest ()
```

### Example 17
[Code]
from collections import Counter

class Solution :
    def sumOfUnique ( self , nums : List [ int ]) -> int :
        return sum ( k for k , v in Counter ( nums ) . items () if v == 1)

[Property]
category: function implementation
language: Python
difficulty: easy
concepts: dictionary -based counting , filtering unique elements with
    list comprehension , summation of elements based on condition

[Instruction]
Write a Python function that takes a list from user input and prints
    the sum of counts for elements that occur more than once. Use a
    dictionary to count frequencies efficiently. Utilize a list
    comprehension to extract repeated elements based on the frequency
    counts , and sum their corresponding counts from the dictionary.

[Response]
To implement this function , we first need to parse the user input as a
     list of integers and then count the frequency of each element
    using a dictionary. We can then calculate the sum of the counts
    for elements that occur more than once.

Here 's a Python function that accomplishes this:

```python
def sum_repeated_counts ():
    nums = list ( map ( int , input () . split ()))
    freq_dict = {}
    for num in nums :
        freq_dict [ num ] = freq_dict . get ( num , 0) + 1
    sum_of_repeats = sum ( v for k , v in freq_dict . items () if v > 1)
    print ( sum_of_repeats )
```

[Tests]
We can test this program with I/O redirection to simulate user input
    and output. Here is an example:

```python
import sys
from io import StringIO
```

```
test_cases = [
    ("1 2 2 3 3 3\n", "5\n"),
    ("1 1 1 1 1\n", "5\n"),
    ("1 2 3 4 5\n", "0\n"),
    ("\n", "0\n")
]

for input_data, expected_output in test_cases:
    stdin, stdout = sys.stdin, sys.stdout
    sys.stdin = StringIO(input_data)
    sys.stdout = StringIO()
    sum_repeated_counts()
    output = sys.stdout.getvalue()
    assert output == expected_output
```

### Example 18

[Code]
```
fp=open('cmd.sh','r')
lines=fp.readlines()
for i,line in enumerate(lines):
    cmd=line.replace('\n','')
    print i,cmd
    p=os.popen(cmd)
    x=p.read()
    p.close()
fp.close
```

[Property]
category: program implementation
language: Python
difficulty: medium
concepts: iterating over file lines, command execution from file input
    , new line character removal

[Instruction]
Develop a Python program that reads a text file containing one shell
    command per line and executes these commands sequentially. Each
    command should be sanitized by removing unnecessary newline
    characters before execution. Provide clear error handling and
    logging to manage exceptions that might occur during the execution
     of the commands.

[Response]
To create a Python program that reads and executes shell commands from
     a text file, we can use Python's built-in module `subprocess` for
     running the commands and `logging` for handling logs and errors.
    The program will involve reading and sanitizing the commands from
    a file, executing them sequentially, and logging the results or
    errors.

Below is the Python script that demonstrates this:

```python
import subprocess
import logging

logging.basicConfig(level=logging.INFO, format='%(asctime)s:%(
    levelname)s:%(message)s')

def read_commands(file_path):
    """Reads a text file and returns a list of commands."""
    try:
```

```python
        with open(file_path, 'r') as file:
            commands = file.readlines()
        commands = [cmd.strip() for cmd in commands]  # Remove newline
 characters and spaces
        return commands
    except FileNotFoundError:
        logging.error(f"The file {file_path} was not found.")
        raise
    except Exception as e:
        logging.error(f"An error occurred while reading the file: {str
(e)}")
        raise

def execute_command(command):
    """Executes a single shell command."""
    try:
        logging.info(f"Executing command: {command}")
        result = subprocess.run(command, shell=True, check=True,
    stdout=subprocess.PIPE, stderr=subprocess.PIPE)
        logging.info(f"Command output: {result.stdout.decode().strip()
    }")
    except subprocess.CalledProcessError as e:
        logging.error(f"Error executing command '{command}': {e.stderr
.decode()}")
    except Exception as e:
        logging.error(f"An unexpected error occurred: {str(e)}")

def main(file_path):
    """Main function to read commands from a file and execute them."""
    commands = read_commands(file_path)
    for command in commands:
        execute_command(command)
```

[Tests]
To test this program, we can use Python's `unittest` framework along
    with `mock_open` to simulate file operations, `patch` to mock `
    subprocess.run` for testing command execution, and `assertLogs` to
     verify the logging output. Here's an example of how you can write
     tests for this program:

```python
import unittest
from unittest.mock import mock_open, patch, MagicMock

class TestCommandExecutor(unittest.TestCase):
    def test_read_commands_success(self):
        m = mock_open(read_data="ls\npwd\n")
        with patch('builtins.open', m):
            result = read_commands("dummy_path")
            self.assertEqual(result, ['ls', 'pwd'])

    def test_read_commands_file_not_found(self):
        with patch('builtins.open', mock_open()) as mocked_file:
            mocked_file.side_effect = FileNotFoundError
            with self.assertRaises(FileNotFoundError):
                read_commands("nonexistent_path")

    @patch('subprocess.run')
    def test_execute_command_success(self, mock_run):
        mock_run.return_value = MagicMock(stdout=b"Execution
    successful", stderr=b"")
        execute_command("ls")
        mock_run.assert_called_once_with("ls", shell=True, check=True,
     stdout=subprocess.PIPE, stderr=subprocess.PIPE)
```

```python
    @patch('subprocess.run')
    def test_execute_command_failure(self, mock_run):
        error_message = b"Command failed"
        mock_run.side_effect = subprocess.CalledProcessError(1, 'ls',
    stderr=error_message)
        with self.assertLogs(level='ERROR') as log:
            execute_command("ls")
            self.assertIn("Error executing command 'ls'", log.output
    [0])
            self.assertIn("Command failed", log.output[0])

if __name__ == "__main__":
    unittest.main()
```

### Example 19
[Code]
```
def busca_linear_recursiva(array,elemento):
    i = -1
    return busca_recursiva(array,elemento, i)

def busca_recursiva(array,elemento, i):
    i += 1
    if(i == len(array)):
        return -1
    if(array[i] == elemento):
        return i
    return busca_recursiva(array,elemento,i)
```

[Property]
category: program implementation
language: Python
difficulty: medium
concepts: linear search with recursion, default parameter
    initialization, multiple base cases in recursion

[Instruction]
Design a Python program that not only searches for a target value in a
    list using recursion but also constructs a trace of the indices
    it checks. This program should return both the index of the target
    , if found, or `-1` if not, alongside a list of the indices it
    examined during the search. The function should handle cases where
     the target is not present in the list.

[Response]
We can implement a recursive linear search function that returns the
    index of the target element if found or `-1` if not. To keep track
     of the indices examined during the search, we can pass a list as
    an additional parameter to store these indices.

Here is the Python program that implements this functionality:

```python
def search_recursive(arr, target, index, indices_checked):
    """Recursive function to search for the target value in the list
    ."""
    if index >= len(arr):
        return -1
    indices_checked.append(index)
    if arr[index] == target:
        return index
    return search_recursive(arr, target, index + 1, indices_checked)

def recursive_search(arr, target):
```

```
    """Wrapper function to initiate recursive search."""
    indices_checked = []
    result = search_recursive(arr, target, 0, indices_checked)
    return result, indices_checked

# Example usage
if __name__ == "__main__":
    array = [2, 3, 4, 8, 5, 9, 1, 6, 5, 7, 5]
    target = 6
    result, indices = recursive_search(array, target)
    print(f"Target {target} found at index {result} with indices
    checked: {indices}")
```

In this program, the `search_recursive` function recursively searches
    for the target element in the list and appends the indices checked
     to the `indices_checked` list. The `recursive_search` function
    acts as a wrapper to initiate the recursive search and return the
    result along with the list of indices checked.

[Tests]
To test the implementation, we need to ensure that both the index of
    the target element and the list of indices checked are correct.
    Here are the test cases:

```python
# Test case 1: Target found in the list
arr1 = [2, 3, 4, 8, 5, 9, 1, 6, 5, 7, 5]
target1 = 6
expected_result1 = arr1.index(target1)
expected_indices1 = list(range(expected_result1 + 1))
result1, indices_checked1 = recursive_search(arr1, target1)
assert result1 == expected_result1
assert indices_checked1 == expected_indices1

# Test case 2: Target not found in the list
arr2 = [1, 2, 3, 4, 5]
target2 = 6
expected_result2 = -1
expected_indices2 = list(range(len(arr2)))
result2, indices_checked2 = recursive_search(arr2, target2)
assert result2 == expected_result2
assert indices_checked2 == expected_indices2

# Test case 3: Target found at the end of the list
arr3 = [5, 4, 3, 2, 1]
target3 = 1
expected_result3 = len(arr3) - 1
expected_indices3 = list(range(expected_result3 + 1))
result3, indices_checked3 = recursive_search(arr3, target3)
assert result3 == expected_result3
assert indices_checked3 == expected_indices3
```

### Example 20
[Code]
Data = np.zeros([break_points[-1],size_blocks])
Data_stacked = np.zeros([break_points[-1]-num_blocks+1, size_blocks*
    num_blocks])
cluster_point_list = []
for counter in xrange(len(break_points)):
        break_pt = break_points[counter]
        cluster = seg_ids[counter]
        if counter == 0:
                old_break_pt = 0
```

```python
        else:
                old_break_pt = break_points[counter-1]
        for num in xrange(old_break_pt,break_pt):
                ##generate the point from this cluster
                # print "num is:", num
                if num == 0:
```

[Property]
category: program implementation
language: Python
difficulty: medium
concepts: NumPy array initialization, iteration over break points,
    cluster segmentation, nested loops

[Instruction]
Implement a Python program that initializes a NumPy array randomly
    (0--1) and segments it into clusters based on provided breakpoints
    . Each cluster is defined by consecutive elements between two
    breakpoints. The output should be a list of NumPy arrays
    representing these clusters.

For example, given a size 10 and breakpoints `[3, 6, 8]`, the program
    should first initialize a NumPy array of 10 random elements
    between 0 and 1 and then output three clusters: elements at
    indices `0-2`, `3-5`, `6-7`, and `8-9`. Ensure that the program
    handles cases where breakpoints are at the boundaries of the array
     or if there are no breakpoints, in which case the entire array
    should be a single cluster. Include error handling to manage
    invalid inputs such as negative array sizes or breakpoints that
    are out of range.

[Response]
Here is the program that achieves what you've described:

```python
import numpy as np

def segment_array(data, break_points):
    """Segment the given array into clusters based on specified
    breakpoints."""
    break_points = [0] + break_points + [len(data)]
    segmented_arrays = []
    for i in range(len(break_points) - 1):
        start_index = break_points[i]
        end_index = break_points[i + 1]
        segmented_arrays.append(data[start_index:end_index])
    return segmented_arrays

def create_clusters(size, break_points):
    """Generate and segment a random array of given size using
    breakpoints."""
    if size <= 0:
        raise ValueError("Array size must be positive.")
    data = np.random.rand(size)
    clusters = segment_array(data, break_points)
    return clusters

# Example usage
if __name__ == "__main__":
    size = 10
    break_points = [3, 6, 8]
    clusters = create_clusters(size, break_points)
    for i, cluster in enumerate(clusters):
        print(f"Cluster {i}: {cluster}")
```

In this implementation, `segment_array` transforms the input data
    array into a list of NumPy arrays based on the provided
    breakpoints. The `main` function initializes a random NumPy array
    of a specified size and calls `segment_array` to segment the array
     into clusters based on the breakpoints.

[Tests]
To test the program, we can verify that the values in each cluster are
     within the expected range (0--1) and that the clusters are
    correctly segmented. Here are some test cases:

```python
def test_segmentation(size, break_points):
    clusters = create_clusters(size, break_points)
    for cluster in clusters:
        assert np.all(cluster >= 0) and np.all(cluster <= 1)
    break_points = [0] + break_points + [size]
    for i in range(len(break_points) - 1):
        assert len(clusters[i]) == break_points[i + 1] - break_points[
    i]

# Test cases
test_cases = [
    (20, [5, 10, 15]),
    (10, [3, 6, 8]),
    (15, [3, 6, 12]),
    (30, [7, 14, 21, 27]),
    (5, [1]),
    (10, []),
    (50, [10, 20, 30, 40]),
]

for size, breakpoints in test_cases:
    test_segmentation(size, breakpoints)
```

### Example 21
[Code]
if match_case == 'auto':
    if len(set([ch_name.lower() for ch_name in ch_names])) < len(set(
    ch_names)):
        match_case = True
    else:
        match_case = False

if match_case:
    picks = mne.pick_channels(ch_names, pick_chs, ordered=ordered)
else:
    ch_names = [ch_name.lower() for ch_name in ch_names]
    pick_chs = [pick_ch.lower() for pick_ch in pick_chs]
    picks = mne.pick_channels(ch_names, pick_chs, ordered=ordered)

return picks

[Property]
category: program implementation
language: Python
difficulty: medium
concepts: case-insensitive string matching, controlled case matching
    based on an option, list comprehension

[Instruction]
Construct a Python program that processes a list of strings provided
    by the user. The first element is a target string, the second is

either 'CaseSensitive' or 'CaseInsensitive' to determine the
matching approach, and the remaining elements are candidates for
comparison. The program should output the number of times the
target appears among the candidates according to the specified
matching rule. The solution should include robust error handling
to validate the matching rule and check that at least two strings
are provided before proceeding.

For example, given the input `['apple', 'CaseInsensitive', 'Apple', '
banana', 'APPLE']`, the program should output `2` since 'Apple'
and 'APPLE' match the target 'apple' in a case-insensitive manner.

[Response]
Here's a Python program that fulfills the requirements you specified.
It first parses the input list to validate the presence of the
target string and matching rule. Then, it compares the target with
the candidate strings, adjusting for case sensitivity as
specified. Finally, it outputs the count of matches.

```python
def count_matches(input_list):
    if not input_list or len(input_list) < 2:
        raise ValueError("Input list must contain at least two items:
the target string and the matching rule.")

    target_string, matching_rule = input_list[0], input_list[1]
    if matching_rule not in ['CaseSensitive', 'CaseInsensitive']:
        raise ValueError("Matching rule must be either 'CaseSensitive'
 or 'CaseInsensitive'.")

    candidates = input_list[2:]
    if matching_rule == 'CaseSensitive':
        match_count = sum(1 for candidate in candidates if candidate
== target_string)
    else:
        target_string = target_string.lower()
        match_count = sum(1 for candidate in candidates if candidate.
lower() == target_string)

    return match_count

def main():
    input_list = input().split()
    print(count_matches(input_list))

if __name__ == "__main__":
    main()
```

The `count_matches` function implements the main logic of the program,
 counting the occurrences of the target string among the candidate
 strings based on the specified matching rule. The `main` function
 reads the input list from the user, calls `count_matches`, and
prints the count of matches.

[Tests]
You can test the program with various inputs to ensure it handles
different scenarios correctly:

```python
assert count_matches(['apple', 'CaseInsensitive', 'Apple', 'banana', '
APPLE']) == 2
assert count_matches(['apple', 'CaseSensitive', 'Apple', 'banana', '
APPLE']) == 0
assert count_matches(['apple', 'CaseInsensitive']) == 0
```

```
assert count_matches(['apple', 'CaseInsensitive', 'apple', 'APPLE', '
    aPpLe', 'banana']) == 3
assert count_matches(['apple', 'CaseSensitive', 'apple', 'APPLE', '
    aPpLe', 'banana']) == 1

try:
    count_matches(['apple'])
    assert False
except ValueError as e:
    assert str(e) == "Input list must contain at least two items: the
    target string and the matching rule."

try:
    count_matches(['apple', 'InvalidRule', 'Apple'])
    assert False
except ValueError as e:
    assert str(e) == "Matching rule must be either 'CaseSensitive' or
    'CaseInsensitive'."
```

