# OpenReview forum: "SelfCodeAlign: Self-Alignment for Code Generation"
_NeurIPS.cc/2024/Conference — NeurIPS 2024 poster_

### Official Review · Reviewer_4fJs · 2024-06-29

**Soundness:** 3
**Presentation:** 3
**Contribution:** 3
**Rating:** 7
**Confidence:** 3

**Summary:**

- The paper introduces SelfCodeAlign, a fully transparent and permissive self-alignment pipeline for code generation in LLMs without relying on extensive human annotations or distillation from larger models. SelfCodeAlign generates instruction-response pairs from seed snippets, evaluates responses with test cases, and fine-tunes models based on successful executions. The approach shows superior performance over state-of-the-art methods, including GPT-3.5-Turbo-based distillation, particularly in HumanEval+ benchmark. The pipeline demonstrates effectiveness across various model sizes, emphasizing the benefits of self-generated data over teacher models with smaller performance gaps.
- Overall, I feel that SelfCodeAlign is a very easy workflow to follow and I see much potential for such pipelines that do not depend on distillation or human annotations. I recommend an accept.

**Strengths:**

## Originality
- The paper adequately cites related work, clearly identifying gaps such as the lack of transparency in existing methods, which is a key motivation for their work.

## Quality
- The submission is technically sound with both quantitative and qualitative analysis.
- The authors provide detailed experimental results, demonstrating significant performance improvements over baselines.
- The inclusion of both large-scale and small-scale model evaluations further strengthens the quality of the research.
- In terms of ethical considerations, they have considered all terms of use as well as the data in code snippets.

## Clarity
- Well organized paper, except for appendix.

## Significance
- The results are highly significant, as SelfCodeAlign achieves performance improvements, notably surpassing models that are an order of magnitude larger in size. This work addresses the challenge of instruction tuning without human annotations or distillation, offering a scalable and transparent solution that advances the state of the art in code generation.

**Weaknesses:**

## Originality
- Perhaps similar to this paper [Beyond Human Data: Scaling Self-Training for Problem-Solving with Language Models](https://arxiv.org/pdf/2312.06585) ? Even if it is different, I think that this should also be part of your baseline comparison as well.

## Quality
- The qualitative examples provided in the appendix are excessively long, which may overwhelm the reader and obscure the main differences and contributions of the methodology. It would be beneficial to reduce the number of examples or to shorten them, focusing on highlighting the key differences and improvements over baseline methods. Additionally, the examples are presented in black and white with no descriptions or annotations, making it difficult to discern their significance. Providing clearer, annotated examples with concise explanations would enhance the readability and impact of this section.
- I do not see any weaknesses discussed in this work, for example, in what scenario do you think does this methodology not work? Why is the score still not perfect? (or for eg, below 80% accuracy)

**Questions:**

- What about experiments/benchmarking on models that uses the GPT4 family as part of distillation?
- Why did you only limit it to 33B, what about 70b?
- Line 121: What is the difficulty for? It is subjective, so how does the difficulty aid the model/aid in finetuning of models?
- Line 228-233, Table 7: Any reason why the increasing trend is not consistent, for eg for StarCoder2-15B, the score decreased when tuned with DeepSeek-Coder-33B data?
- Line 232, Table 7: Why did you not fill up the blank cells just like the last row? This would have ensured that your statement is true for all models, because you are just basing off CodeQwen1.5-7B model only.

**Limitations:**

Limitations are stated. However, I think the authors miss one important point – the reliance on the seed snippets. The performance of the model is based on what it was finetuned on, so if the seed snippets are not sufficiently diverse or representative of the target tasks, then it might have resulted in a large drop in accuracy.

---

> ### Author Rebuttal · Authors · 2024-08-07
>
> Thank you for your comprehensive review and important suggestions! Also thanks for pointing out the presentation issues in Appendix, which we will fix in the revision. We provide our responses to your questions and concerns as follows.
>
> > Q1: What about experiments/benchmarking on models that uses the GPT4 family as part of distillation?
>
> Please kindly note that we included two GPT-4-distilled models in Table 1: OpenCodeInterpreter-DS-33B [1] and Magicoder-S-DS-6.7B [2]. We also conducted an experiment directly distilling GPT-4 responses to our 74k SelfCodeAlign dataset. As shown in the table below, GPT-4o still generates 14% erroneous code, and training on the dataset produced by SelfCodeAlign outperformed GPT-4o distillation.
>
> |Teacher Model|Execution Pass Rate|HumanEval+|
> |-|-|-|
> |GPT-4o|86%|65.9|
> |CodeQwen1.5-7B|100%|67.1|
>
> [1] Zheng et al. OpenCodeInterpreter: Integrating Code Generation with Execution and Refinement.
>
> [2] Wei et al. Magicoder: Empowering Code Generation with OSS-Instruct.
>
> > Q2: Why did you only limit it to 33B, what about 70b?
>
> Thank you for your suggestion. Due to resource constraints, we were only able to experiment with the 33B model as the largest. We plan to explore larger models in future work as we acquire additional resources.
>
> > Q3: Line 121: What is the difficulty for? It is subjective, so how does the difficulty aid the model/aid in finetuning of models?
>
> Great question. The difficulty attribute is randomly sampled to create problems of varying complexity, which is important for code instruction-tuning datasets according to prior research [3]. To demonstrate the effectiveness of the "difficulty" attribute, we analyzed the 74k SelfCodeAlign dataset. As shown in the table below, the decreasing pass@1 rate as difficulty increases shows that the attribute is meaningful. SelfCodeAlign generates 10 responses for each instruction and keeps any that pass. This approach leads to higher overall success rates (pass@10) and more similar rates across difficulty levels, creating a more balanced final dataset.
>
> |Difficulty|Pass@1|Pass@10|Number of Samples|
> |-|-|-|-|
> |Easy|30.8|77.2|25.6k
> |Medium|27.9|75.9|24.6k|
> |Hard|25.5|74.6|23.7k|
>
> [3] Luo et al. WizardCoder: Empowering Code Large Language Models with Evol-Instruct.
>
> > Q4: Line 228-233, Table 7: Any reason why the increasing trend is not consistent, for eg for StarCoder2-15B, the score decreased when tuned with DeepSeek-Coder-33B data?
>
> The inconsistent trend aligns with our observation in lines 226-228: when the performance gap between models is small, a base model may benefit more from self-generated data than from data generated by a slightly stronger teacher. While DeepSeek-Coder-33B has a higher HumanEval score, StarCoder2-15B outperforms it on math and code reasoning benchmarks [4], indicating their overall performance gap is not substantial.
>
> [4] Lozhkov et al. StarCoder 2 and The Stack v2: The Next Generation.
>
> > Q5: Line 232, Table 7: Why did you not fill up the blank cells just like the last row?...
>
> Good point. We intentionally left those cells blank. Most models in Table 7 were evaluated using both self-generated and stronger-model-generated data to demonstrate self-alignment effectiveness and show distribution shift impact. The last row is an exception, as it represents our strongest base model, which doesn't have a "stronger" model to generate data from.
>
> > C1: Perhaps similar to this paper Beyond Human Data: Scaling Self-Training for Problem-Solving with Language Models…
>
> Great suggestion. We'll discuss this paper in our revision. The key difference is that SelfCodeAlign expands and improves both input and output spaces systematically, while "Beyond Human Data" focuses solely on gathering high-quality outputs from a fixed original dataset. This highlights our approach's broader scope in generating diverse coding problems and solutions.
>
> > C2: The qualitative examples provided in the appendix are excessively long…
>
> We appreciate your feedback. We will improve our presentation and make all the examples concise, clear, and easy to comprehend in our revised manuscript.
>
> > C3: I do not see any weaknesses discussed in this work…for example, in what scenario do you think does this methodology not work? Why is the score still not perfect?
>
> Thank you for noting this. We did discuss limitations in Section 6, but we can expand further. The main weakness of the methodology is its reliance on test execution. In practice, not all solutions can be verified through execution, and generated tests can also be erroneous. These factors contribute to imperfect scores.
>
> > C4: ...if the seed snippets are not sufficiently diverse or representative of the target tasks, then it might have resulted in a large drop in accuracy
>
> This is an excellent comment. We ensure seed quality and diversity through rigorous mining, filtering, and deduplication, which we describe in Appendix A. We will further elaborate on this point in our revision.

---

### Official Review · Reviewer_pw5h · 2024-07-12

**Soundness:** 2
**Presentation:** 2
**Contribution:** 3
**Rating:** 6
**Confidence:** 2

**Summary:**

The authors proposed SelfCodeAlign that finetunes the model based on the filtered data generated by the same model itself. The authors conduct experiments to show that SelfCodeAlign outperforms most open-sourced models that were finetuned on public code dataset.

**Strengths:**

The code generation problem is important and the results (compared to models trained on public dataset) are promising.

**Weaknesses:**

Compare to models that are distilled/trained on non-disclosed data, the performance of SelfCodeAlign is not as competitive. The presentation can be improved, see **questions**.

**Questions:**

1. Can you properly highlight the row in table 1?
2. I would suggest to give a brief summarization of the component analysis after line 92.
3. It seems crucial to me to understand why SelfCodeAlign outperforms other dataset (for example, GPT-generated dataset), is there an analysis on how these datasets differ? Also, you mentioned distribution shift across different models in 4.1, is there a qualitative/quantitative comparison between the code generated by different models?

**Limitations:**

Limitations have been discussed.

---

> ### Author Rebuttal · Authors · 2024-08-07
>
> Thank you for your valuable review and suggestions! We provide our response as follows.
>
> > Q1: Can you properly highlight the row in table 1?
>
> Thanks for the feedback. We appreciate your suggestions for improving Table 1. Could you kindly provide more specific details regarding your concerns, such as which rows you believe need to be highlighted? This will help us make the necessary adjustments more effectively.
>
> > Q2: I would suggest to give a brief summarization of the component analysis after line 92.
>
> Absolutely. We will ensure to include the summarization in our revised manuscript.
>
> > Q3: …why SelfCodeAlign outperforms other dataset (for example, GPT-generated dataset), is there an analysis on how these datasets differ? Also, you mentioned distribution shift … is there a qualitative/quantitative comparison between the code generated by different models?
>
> Great questions. Regarding your first question, as shown in Section 4.2, execution filtering and code correctness are important to the effectiveness of self-alignment. The Evol-Instruct [1] and OSS-Instruct [2] datasets used in the paper are direct distillation from GPT-3.5-Turbo, which means they may include incorrect code that harms the model. To verify this, we use the GPT-4o model to generate responses for the 74k SelfCodeAlign dataset and compare it with the original dataset, keeping the instructions the same. The table below shows that while GPT-4o is stronger, it still generates 14% erroneous code, and using its outputs for instruction tuning is less effective than using execution-filtered outputs from CodeQwen1.5-7B:
>
> |Teacher Model|Execution Pass Rate|HumanEval+|
> |-|-|-|
> |GPT-4o|86%|65.9|
> |CodeQwen1.5-7B|100%|67.1|
>
> Second, Section 4.1 indicates that a base model can benefit more from data within its own distribution than a shifted teacher distribution. In the table below, we compare the perplexity of our base model, CodeQwen1.5-7B, on self-generated data and two GPT-generated datasets. It shows that CodeQwen1.5-7B has a lower perplexity on self-generated data, suggesting it is easier to learn from. This observation complies with the finding from [3] that self-generated positive data with lower perplexity yields better finetuning performance.
>
> |Dataset|Perplexity|
> |-|-|
> |SelfCodeAlign|2.12|
> |OSS-Instruct|2.20|
> |Evol-Instruct|3.76|
>
> For your second question, we compute the perplexity of CodeQwen1.5-7B on the outputs from different models to ​​quantitatively measure the distribution shift. The table below shows that self-generated data achieves the lowest perplexity:
>
> |Model that Generates Data|Perplexity|
> |-|-|
> |StarCoder2-3B|2.67|
> |Llama3-8B|2.73|
> |StarCoder2-15B|2.28|
> |DeepSeek-Coder-33B|2.25|
> |CodeQwen1.5-7B|2.13|
>
> Please also kindly refer to Appendix C.2, which provides qualitative examples of the outputs generated by different models at each step of the SelfCodeAlign framework.
>
> [1] Luo et al. WizardCoder: Empowering Code Large Language Models with Evol-Instruct.
>
> [2] Wei et al. Magicoder: Empowering Code Generation with OSS-Instruct.
>
> [3] Setlur et al. RL on Incorrect Synthetic Data Scales the Efficiency of LLM Math Reasoning by Eight-Fold.
>
> > C1: Compare to models that are distilled/trained on non-disclosed data, the performance of SelfCodeAlign is not as competitive.
>
> We want to kindly highlight that SelfCodeAlign still excels over many larger, proprietary models, including Gemini Pro, Mistral Large, and CodeLlama-70B-Instruct. As also mentioned by Reviewer 4fJs, “The results are highly significant, as SelfCodeAlign achieves performance improvements, notably surpassing models that are an order of magnitude larger in size”. Additionally, in Q3, we demonstrate that our approach outperforms direct distillation from the much stronger GPT-4o model.
>
> Meanwhile, SelfcodeAlign is the first fully transparent and permissive pipeline for self-aligning code LLMs without extensive human annotations or distillation. We are more than happy to further discuss this point during the discussion period.

---

> > ### Comment · Reviewer_pw5h · 2024-08-12
> >
> > Thanks for the response, my concerns are addressed, and I am raising my score.

---

> > > ### Author Response · Authors · 2024-08-12
> > >
> > > Thank you for taking the time to read our response. We truly appreciate it! Should you have any new questions or concerns, please don't hesitate to let us know.

---

### Official Review · Reviewer_qGfX · 2024-07-13

**Soundness:** 3
**Presentation:** 3
**Contribution:** 2
**Rating:** 5
**Confidence:** 3

**Summary:**

This paper introduces SelfCodeAlign, an entirely transparent and permissive pipeline designed for self-aligning code large language models without the need for human annotations or distillation. By applying SelfCodeAlign to CodeQwen1.5-7B, the authors generated a dataset containing 74k instruction-response pairs. They then fine-tuned CodeQwen1.5-7B using this dataset, resulting in SelfCodeAlign-CQ-7B, which demonstrates robust performance on the HumanEval+ benchmark.

**Strengths:**

1. The performance is satisfactory: SelfCodeAlign-CQ-7B achieves a pass@1 score of 67.1 on HumanEval+, outperforming larger models like CodeLlama-70B-Instruct (65.2), which is a significant achievement.
2. The process is auto-mated: This paper introduces a novel self-alignment pipeline including concept extraction from seed code, task generation, multiple response generation, and execution validation. This approach is independent of human annotations or large model distillation, making it easy to be applied.
3. Scalability: Experiments demonstrate the method's applicability to models ranging from 3B to 33B parameters, showing good scalability across different model sizes.

**Weaknesses:**

1. Lack of Diversity in Generated Tasks: While the method aims to produce a variety of coding tasks, it is unclear how this diversity is achieved or measured. There is a risk that the generated tasks may be biased towards certain types of coding problems, which could limit the model's ability to generalize effectively.
2. Overreliance on Self-Generated Tests: The method relies heavily on tests generated by the model itself to validate responses. This self-validation approach could result in a feedback loop where the model learns to create tests that are easy to pass, rather than generating truly challenging or comprehensive tests. The paper does not address how this potential issue is mitigated.

**Questions:**

Refer to the weakness.

**Limitations:**

Refer to the weakness.

---

> ### Author Rebuttal · Authors · 2024-08-07
>
> Thank you for your comprehensive review and suggestions! We address your questions as follows.
>
> > Q1: Lack of Diversity in Generated Tasks: While the method aims to produce a variety of coding tasks, it is unclear how this diversity is achieved or measured…
>
> Good question. We ensure task diversity through a rigorous seed gathering pipeline (Appendix A) with quality filtering and deduplication, and 21 carefully designed, diverse few-shot examples (Appendix D) to enhance response generation variety. To measure diversity, we conducted comprehensive evaluations across 7 coding benchmarks covering 4 different problem types. The evaluation results consistently demonstrate SelfCodeAlign's effectiveness in different coding tasks.
>
> > Q2: Overreliance on Self-Generated Tests: …This self-validation approach could result in a feedback loop where the model learns to create tests that are easy to pass, rather than generating truly challenging or comprehensive tests…
>
> This is a great point. We've implemented two key strategies to mitigate the risk of generating overly easy problems. First, during instruction generation, we have an attribute “difficulty” whose value is randomly sampled from easy/medium/hard. Second, for each instruction, we generate 10 different responses and choose any of the passing responses. This ensures that the model is exposed to a variety of problem-solving approaches and not just always the easiest path. In the table below, we analyze the 74k SelfCodeAlign dataset. The decreasing pass@1 rate with increasing difficulty shows the attribute is meaningful, while the higher and more consistent pass@10 rates help to create a balanced dataset:
>
> |Difficulty|Pass@1|Pass@10|Number of Samples|
> |-|-|-|-|
> |Easy|30.8|77.2|25.6k
> |Medium|27.9|75.9|24.6k|
> |Hard|25.5|74.6|23.7k|
>
> We also experimented with directly distilling GPT-4o responses to the SelfCodeAlign dataset. As the table shows, our approach outperforms direct distillation from GPT-4o, despite using a weaker base model. Notably, even GPT-4o generates 14% erroneous code, highlighting the importance of execution-based validation in our method.
>
> |Teacher Model|Execution Pass Rate|HumanEval+|
> |-|-|-|
> |GPT-4o|86%|65.9|
> |CodeQwen1.5-7B|100%|67.1|

---

> > ### Comment · Reviewer_qGfX · 2024-08-13
> >
> > Hi,
> > Thank you for your thoughtful response. My main concern has been addressed, and I'd like to update my score to a 5.

---

> > > ### Author Response · Authors · 2024-08-13
> > >
> > > Thanks for your feedback! We sincerely appreciate the time you took to read our response. If you have any further questions or concerns, please feel free to reach out.

---

### Official Review · Reviewer_NEpV · 2024-07-15

**Soundness:** 3
**Presentation:** 3
**Contribution:** 3
**Rating:** 7
**Confidence:** 3

**Summary:**

This paper proposes a pipeline for generating synthetic instruction tuning data. The method consists of the following steps: 1. data filtering is applied to seed coding data to select high quality examples; 2. base LLM is used to generate a set of coding concept and category based on the seed data; 3. base LLM is used to generate coding instruction, response and test; 4. generated examples are selected based on the code execution result.

**Strengths:**

1. the paper focuses on using base model to generate synthetic data to self-improve, which is an interesting and useful angle for synthetic data generation
2. the method is evaluated on several different coding LLM benchmarks which shows the effectiveness of the method
3. there are also ablation experiments verifying the contribution of specific design choices in the framework.

**Weaknesses:**

While using base model to self-improve is an interesting and useful direction, synthetic data generation could be improved by using a stronger LLM than the base model. It is not clear from the paper whether the proposed framework would be effective compared to previous methods if we use a stronger LLM to synthesize the data. The synthetic data generation could also be potentially improved by having multiple rounds of data generation process.

**Questions:**

1. Have you tried this framework using stronger LLM to generate synthetic data?
2. Can you get even better performance by running several rounds of data generation with improved base model?

**Limitations:**

Yes

---

> ### Author Rebuttal · Authors · 2024-08-07
>
> Thank you for your valuable review and suggestions! We put our responses to your questions as follows.
>
> > Q1: Have you tried this framework using stronger LLM to generate synthetic data?
>
> Thank you for your question. We want to kindly highlight that we explored this point in Section 4.1, which examines the effectiveness of the SelfCodeAlign framework on various base models. Table 7 demonstrates that all models benefit from the framework, and weaker models (e.g., StarCoder2-3B) can achieve larger performance gains when trained on data synthesized by stronger models (e.g., DeepSeek-Coder-33B).
>
> > Q2: Can you get even better performance by running several rounds of data generation with improved base model?
>
> Thank you for your insightful suggestion. We agree that an improved base model could potentially produce better synthetic data. While time constraints during the rebuttal period prevent us from conducting additional experiments, we appreciate your point and plan to incorporate this idea with relevant experiments in our final revision.

---

### Author Rebuttal · Authors · 2024-08-07

We deeply appreciate all the reviewers for their insightful feedback and suggestions for our work. In our responses below, we address each primary question (denoted as Q) or comment (denoted as C) raised by the individual reviewers. Additionally, we will revise our paper to incorporate editorial suggestions. Should there be any misunderstandings of the questions, please kindly let us know; we are eager to communicate with all the reviewers throughout the discussion period.

---

### Decision · Program_Chairs · 2024-09-25

**Decision:**

Accept (poster)

**Comment:**

The paper studies a method that generates synthetic data for instruction finetuning Code LLMs. They first generate a diverse set of data examples, convert and validate them. The authors are able to train a 7B model to match the performance of CodeLlama-70B-Instruct on HumanEval+. The authors addressed the questions and concerns from the reviewers and discussed key design choices of the method. The authors also acknowledged the limitations of generated tests and execution verifications.